# Mathematical modeling of plus-strand RNA virus replication to identify broad-spectrum antiviral treatment strategies

**Carolin Zitzmann**[1,2]\*, **Christopher Dächert**[3¤a], **Bianca Schmid**[4], **Hilde van der Schaar**[5¤b], **Martijn van Hemert**[6], **Alan S. Perelson**[2], **Frank J. M. van Kuppeveld**[5], **Ralf Bartenschlager**[5,7,8], **Marco Binder**[3], **Lars Kaderali**[1]\*

**1** Institute of Bioinformatics, University Medicine Greifswald, Greifswald, Germany, **2** Theoretical Biology and Biophysics, Los Alamos National Laboratory, Los Alamos, New Mexico, United States of America, **3** Research Group "Dynamics of Early Viral Infection and the Innate Antiviral Response", Division Virus-Associated Carcinogenesis (F170), German Cancer Research Center (DKFZ), Heidelberg, Germany, **4** Dept of Infectious Diseases, Molecular Virology, Heidelberg University, Heidelberg, Germany, **5** Division of infectious Diseases and Immunology, Virology Section, Dept of Biomolecular Health Sciences, Utrecht University, Utrecht, The Netherlands, **6** Department of Medical Microbiology, Leiden University Medical Center, Leiden, The Netherlands, **7** Division Virus-Associated Carcinogenesis (F170), German Cancer Research Center (DKFZ), Heidelberg, Germany, **8** German Center for Infection Research (DZIF), Heidelberg partner site, Heidelberg, Germany

¤a Current address: Max von Pettenkofer Institute, Ludwig-Maximilians-University München, Germany
¤b Current address: VectorY Therapeutics, Amsterdam, The Netherlands
\* czitzmann@lanl.gov (CZ); lars.kaderali@uni-greifswald.de (LK)

**Data Availability Statement:** Data are provided as supplementary material. Code and models underlying the manuscript are available via GitHub

## Abstract

Plus-strand RNA viruses are the largest group of viruses. Many are human pathogens that inflict a socio-economic burden. Interestingly, plus-strand RNA viruses share remarkable similarities in their replication. A hallmark of plus-strand RNA viruses is the remodeling of intracellular membranes to establish replication organelles (so-called "replication factories"), which provide a protected environment for the replicase complex, consisting of the viral genome and proteins necessary for viral RNA synthesis. In the current study, we investigate pan-viral similarities and virus-specific differences in the life cycle of this highly relevant group of viruses. We first measured the kinetics of viral RNA, viral protein, and infectious virus particle production of hepatitis C virus (HCV), dengue virus (DENV), and coxsackie-virus B3 (CVB3) in the immuno-compromised Huh7 cell line and thus without perturbations by an intrinsic immune response. Based on these measurements, we developed a detailed mathematical model of the replication of HCV, DENV, and CVB3 and showed that only small virus-specific changes in the model were necessary to describe the in vitro dynamics of the different viruses. Our model correctly predicted virus-specific mechanisms such as host cell translation shut off and different kinetics of replication organelles. Further, our model suggests that the ability to suppress or shut down host cell mRNA translation may be a key factor for in vitro replication efficiency, which may determine acute self-limited or chronic infection. We further analyzed potential broad-spectrum antiviral treatment options in silico and found that targeting viral RNA translation, such as polyprotein cleavage and viral RNA synthesis, may be the most promising drug targets for all plus-strand RNA viruses. Moreover, we found that targeting only the formation of replicase complexes did not stop the

at https://github.com/Carolin1901/Plus_Strand_RNA_Virus_Replication.

**Funding:** This work received funding from the BMBF through the ERASysAPP project SysVirDrug (grant 031A602A). LK received funding from the DFG (grant number KA 2989/13-1). CD was supported by a stipend of the DKFZ International PhD Program. Portions of this work were done under the auspices of the U.S. Department of Energy under contract 89233218CNA000001 and supported by NIH grants R01-AI078881 and R01-AI116868 to ASP. The funders had no role in study design, data collection and analysis, decision to publish, or preparation of the manuscript.

**Competing interests:** The authors have declared that no competing interests exist.

*in vitro* viral replication early in infection, while inhibiting intracellular trafficking processes may even lead to amplified viral growth.

## Author summary

Plus-strand RNA viruses comprise a large group of related and medically relevant viruses. The current global pandemic of COVID-19 caused by the SARS-coronavirus-2 as well as the constant spread of diseases such as dengue and chikungunya fever show the necessity of a comprehensive and precise analysis of plus-strand RNA virus infections. Plus-strand RNA viruses share similarities in their life cycle. To understand their within-host replication strategies, we developed a mathematical model that studies pan-viral similarities and virus-specific differences of three plus-strand RNA viruses, namely hepatitis C, dengue, and coxsackievirus. By fitting our model to *in vitro* data, we found that only small virus-specific variations in the model were required to describe the dynamics of all three viruses. Furthermore, our model predicted that ribosomes involved in viral RNA translation seem to be a key player in plus-strand RNA replication efficiency, which may determine acute or chronic infection outcomes. Furthermore, our *in-silico* drug treatment analysis suggested that targeting viral proteases involved in polyprotein cleavage, in combination with viral RNA replication may represent promising drug targets with broad-spectrum antiviral activity.

## Introduction

Plus-strand RNA viruses are the largest group of human pathogens that cause re-emerging epidemics, as seen with dengue, chikungunya, and Zika virus, as well as global pandemics of acute and chronic infectious diseases such as hepatitis C and the common cold. The current global SARS-coronavirus-2 (SARS-CoV-2) pandemic shows how our lives can become affected by a rapidly spreading plus-strand RNA virus. As of May 2022, more than 500 million cases of SARS-CoV-2 infections have been reported, with over 6 million confirmed deaths [1,2]. While a global pandemic of the current scale clearly causes exceptional socio-economic burdens [3], various other plus-strand RNA viruses also cause significant burdens. For example, in 2013, symptomatic dengue cases in 141 countries caused socio-economic costs of US$ 8.9 billion [4], while the costs of the latest Zika outbreak have been estimated to be US$ 7–18 billion in Latin America and the Caribbean from 2015 to 2017 [5]. Furthermore, between 2014 and 2018, the USA spent around US$ 60 billion on hepatitis C medication, with around US$ 80,000 per patient [6,7].

Treatment options are limited for the majority of plus-strand RNA viruses. While vaccines and vaccine candidates are available for a few viruses, approved direct-acting antiviral drugs are only available against hepatitis C and SARS-CoV-2 [8,9]. Given the high disease burden and socio-economic cost caused by infections with plus-strand RNA viruses, there is an urgent need for broadly acting antiviral drugs. For their development, it is important to study the life cycles and host restriction and dependency factors in detail, not only at the level of each virus individually but also across a group of related viruses, to gain pan-viral insights. The current study investigated the life cycle of plus-strand RNA viruses. The ultimate goal was to reveal commonly effective antiviral strategies and potential therapeutic target processes in the viral life cycle. To do so, we chose three representatives of plus-strand RNA viruses, hepatitis C, dengue, and coxsackievirus B3 (compare Table 1).

**Table 1. Feature comparison of plus-strand RNA viruses.** DMV: double-membrane vesicles, ER: endoplasmic reticulum, NS: non-structural, S: structural.

| | HCV | DENV | CVB3 |
|---|---|---|---|
| *Virus characteristics* | | | |
| **Family** | Flaviviridae [20] | Flaviviridae [20] | Picornaviridae [20] |
| **Genus** | Hepacivirus [20] | Flavivirus [20] | Enterovirus [20] |
| **Transmission** | Human-to-human [20] | Mosquito-to-human [32] | Human-to-human [15] |
| **Tropism** | Hepatocytes [33] | Dendritic cells, monocytes, macrophages [32] | Brain/neuron, cardiac tissue, hepatocytes [15,34,35] |
| **Genome size** | 9.6 kb [33] | 10.7 kb [32] | 7.5 kb [15] |
| **Number of genes/ encoded proteins** | 10 (3 S and 7 NS proteins) [33] | 10 (3 S and 7 NS proteins) [32] | 11 (4 S and 7 NS proteins) [15] |
| **Replication organelle (RO)** | DMV derived from ER [20] | Invaginated vesicles derived from ER [20] | DMV derived from Golgi and ER [20] |
| **Enveloped** | Yes [20] | Yes [20] | No [20] |
| **Host shut-off of RNA translation** | No [24] | Partially [23] | Yes [22] |
| *Disease characteristics* | | | |
| **Infection outcome** | Acute and chronic [36] | Acute [37] | Primary acute (ability of virus persistence) [15,38] |
| Basic reproductive number ($R_0$) | 1–3 (strain dependent) [39] | 5 [40] | 2.5 to 5.5 (range for different enteroviruses [41,42]) |
| **Incubation period** | 2 weeks to 6 months [36] | 4 to 10 days [37] | 5 days [38] |
| **Exponential growth rate** | Measured in human blood: 2.2 per day (doubling time 7.6 hours) [43] | Primary infection measured in human blood: 4.0 per day (doubling time 4.2 hours) [approximated from [44]] | Measured in mouse blood: 4.5 per day (doubling time 3.7 hours) [approximated from [38]] |
| | Measured in chimpanzees: 1st phase: 1.4 per day (doubling time 12 hours) [45] 2nd phase: 0.1 per day (doubling time 7.5 days) [45] | Secondary infection measured in human blood: 4.6 per day (doubling time 3.6 hours) [approximated from [44]] | Measured in mouse heart: 14.5 per day (doubling time 1.1 hours) [approximated from [38]] |
| **Time to reach peak** | Measured in human blood: 21 days [43] | Measured in human blood: 7 days [44] | Measured in mouse blood and heart: 3 days [38] |
| **Peak viral load** | Measured in human and chimpanzee blood: $10^6$ to $10^7$ RNA per mL [43,45,46] | Measured in human blood: $10^9$ to $10^{10}$ RNA per mL [44] | In mouse blood: $10^6$ RNA per mL [38] |
| | Measured in human liver: $10^8$ RNA per g [43] | | In mouse heart: $10^{11}$ to $10^{12}$ RNA per g [38] |
| **RNA clearance** | Individuals with spontaneous clearance: 4.3 per day (RNA half-life 4 hours) [approximated from [47]] | Primary infection measured in human blood: 2.8 per day (RNA half-life 6 hours) [approximated from [44]] | Measured in mouse blood: 0.7 per day (RNA half-life 24 hours) [approximated from [38]] |
| | Otherwise: persistent RNA [47] | Secondary infection measured in human blood: 4.0 per day (RNA half-life 4.2 hours) [approximated from [44]] | Measured in mouse heart: 1st phase: 1.2 per day (RNA half-life 13.4 hours) [approximated from [38]] 2nd phase: 0.05 per day (RNA half-life 14 days) [approximated from [38]] |
| **Infection duration** | Months to Years [36] | 2 to 3 weeks [44] | 2 weeks [48] |

The enveloped blood-borne hepatitis C virus (HCV) is a *Hepacivirus* of the family *Flaviviridae* that causes acute and chronic hepatitis C. An acute infection is typically mild, but once chronic and untreated, may cause life-threatening conditions, including liver cirrhosis and hepatocellular carcinoma. Approximately 70 million people worldwide live with chronic

hepatitis C, with 400,000 related deaths annually [10]. Notably, hepatitis C can be cured in more than 95% of cases with direct-acting antivirals that inhibit viral replication [10].

The re-emerging dengue virus (DENV) is a *Flavivirus* and belongs, similarly to HCV, to the family *Flaviviridae*. Annually, DENV infects 390 million people worldwide, with around 96 million becoming symptomatic. Unlike HCV, DENV is vector-borne and is spread mainly by the mosquitoes of the *Aedes* species. Infection with DENV causes flu-like illness, occasionally with severe complications primarily associated with heterotypic secondary infections (e.g., hemorrhagic fever and shock syndrome) [11]. The clinical manifestation of a DENV infection is closely related to infections with the mosquito-borne chikungunya and Zika virus, leading to frequent misdiagnosis [12].

Coxsackieviruses are members of the genus *Enterovirus* of the family *Picornaviridae*. This genus includes important human pathogens such as poliovirus, enterovirus-A71 (EV-A71), EV-D68, coxsackievirus, and rhinovirus. Enteroviruses cause 10 to 15 million infections every year and therefore belong to the most prevalent pathogens [13]. Enteroviruses cause various diseases, including hand-foot-and-mouth disease, encephalitis, meningitis, and paralysis [14]. Coxsackie B viruses are also known to infect cardiac tissue, leading to viral myocarditis, which can develop into congestive heart failure [15]. In this study, we focus on coxsackievirus B3 (CVB3).

Despite their broad range of clinical manifestations, transmission routes, and tropism (Table 1), plus-strand RNA viruses share remarkable similarities in their replication strategy. By definition, the genome of plus-strand RNA viruses has the polarity of cellular mRNAs. Therefore, after delivery into cells, the genome is translated, giving rise to a polyprotein that must subsequently be cleaved into viral proteins. These proteins induce host cell membrane rearrangements forming replication organelles (ROs). Either within those ROs or on its outer membrane facing the cytosol, viral RNAs are amplified by the viral replicase complex comprising, amongst others, the RNA-dependent RNA polymerase (RdRp). These ROs are thought to hide viral RNAs from the host immune response, thus protecting them from degradation. In addition, the membranous compartment allows the coordinated coupling of the different steps of the viral replication cycle, i.e., RNA translation, RNA replication, and virion assembly [16–19].

However, there are striking differences in the viral life cycles of the three studied viruses. For example, the morphology of the ROs in which the replication takes place differs considerably. While HCV forms double-membrane vesicles (DMV), DENV induces invaginations of host cellular membranes [20]. CVB3 infection first results in single-membrane tubular structures that subsequently transform into DMVs and multilamellar vesicles [21]. Additionally, HCV and DENV, as representatives of *Flaviviridae*, remodel membranes of the rough endoplasmic reticulum (rER), however, the *Picornaviridae* CVB3 uses the ER and Golgi apparatus for its RO formation [20]. Another interesting feature of CVB3 is its ability to trigger a so-called host translational shut-off, leading to increased viral over host RNA translation [22]. Repressed host RNA translation has also been reported for DENV [23]. However, a host shut-off has not been reported for HCV, which instead shows a parallel translation of viral and host cell RNAs, consistent with the predominantly chronic infection caused by this virus [24].

To identify an efficient, broadly active treatment strategy against viral infectious diseases, a comprehensive knowledge of viruses, as well as their exploitive interaction with the host, is of major importance. Mathematical modeling has proven to be a powerful tool to study viral pathogenesis, transmission, and disease progression and has increased our knowledge about therapeutic intervention and vaccination as well as the involvement of the immune system for viruses such as the human immunodeficiency virus (HIV), HCV, influenza A virus, DENV, Zika virus, and SARS-CoV-2 [25–31]. One of the major strengths of mathematical models is their ability to describe and analyze viral replication in a quantitative, dynamic (time-resolved)

framework and to characterize the influence individual parameters have on the ensuing dynamics. These models thus permit much more profound insights into viral replication and antiviral strategies than static, often more qualitative snapshots of host-pathogen interactions.

In the current study, we reproduced the dynamics of the initial post-infection phase of the life cycle of three representative plus-strand RNA viruses, namely HCV, DENV, and CVB3, with one common mathematical model. Using the model, we identified pan-viral similarities and virus-specific differences in the life cycle of plus-strand RNA viruses that are represented by a unique set of model parameters. The inter-viral differences among the plus-strand RNA viruses under investigation have been further analyzed to study how these differences might be related to clinical disease manifestation, particularly with regard to chronic versus acute infections. Our model suggests that the number of ribosomes available for viral RNA translation may be crucial for either acute or chronic infection outcomes. Furthermore, we studied broad-spectrum antiviral treatment options and found that inhibiting viral proteases involved in polyprotein cleavage and RNA synthesis are promising drug targets.

## Methods

### Kinetic experiments and infectivity titers

**HCV infections.**  $2x10^5$ Lunet-CD81$_{high}$ [49] cells per 6-well were seeded in 2 mL 16 hours prior to infection. To ensure simultaneous infection of all cells, cells were kept at 4˚C for 30 min before medium aspiration and inoculation with pre-cooled PEG-precipitated HCV$_{cc}$ (Jc1) [50] at an MOI of 1 at 4˚C for one hour (1 mL per 6-well). The inoculum was removed and cells were covered with 1 ml per well pre-warmed (37˚C) medium and incubated for one hour at 37˚C. Medium was aspirated and cells were treated with an acid wash protocol to remove extracellular vesicles and unbound virus particles: cells were washed with an acidic solution (0.14 M NaCl, 50 mM Glycine/HCl, pH 3.0, 670 μL per 6-well) for three minutes at 37˚C before neutralization with neutralization buffer (0.14 M NaCl, 0.5 M HEPES, pH 7.5, 320 μL per 6-well) and one wash with pre-warmed medium. After that, fresh medium was added. After indicated time-points, total cellular RNA was extracted by phenol-chloroform extraction. Infected cells were washed prior to lysis according to the acid wash protocol described above. After three washing steps with cold 1x PBS, cells were lysed in GITC buffer (700 μL per 6 well) and RNA was extracted as described [51]. A strand-specific RT-qPCR protocol was used to quantify numbers of (+)- and (-)-strand RNA per cell [52]. TCID50 of supernatants was measured and calculated as described previously [50] and converted to PFU/mL.

**CVB3 infections.**  CVB3 wild-type (wt) and CVB3-Rluc, which carries *Renilla luciferase* upstream of the P1 region, were generated as described previously [53]. Subconfluent mono-layers of HuH7 cells, provided by Prof. R. Bartenschlager, were infected with CVB3 wt or CVB3-Rluc at an MOI of 1 for 45 minutes. After removal of the viral inoculum, cells were washed once with PBS and fresh medium (DMEM supplemented with 10% FBS and penicillin and streptomycin) was added. Every hour up to 9 hours post-infection, cells were collected and subjected to various assays. Each assay was performed on three biological replicates. Cells were either frozen together with the medium, after which progeny virus titers were determined by endpoint titration by the method of Reed and Muench and converted to PFU/mL. Another set of cells were lysed in buffer to determine the luciferase activity as a measure of viral protein translation as described previously [53]. Lastly, cells frozen after aspiration of the medium were used for total RNA isolation and quantification of the amount of viral RNA copies per cell with quantitative PCR as described previously [54].

**DENV infections.**  DENV kinetic measurements of intracellular plus-strand RNA and luciferase activity, as well as extracellular infectious virus titers, have been taken from [55]

(raw data provided by the authors). In brief, $2 \times 10^5$ Huh7 cells were infected with DENV reporter virus expressing Renilla luciferase [56] at an MOI of 10. RNA extraction, qRT-PCR, and Renilla luciferase activity were analyzed from cell lysates. RNA was normalized to the 2 h value. Infectivity titers (TCID50/mL) were measured from viral supernatant by limited dilution assays and converted to PFU/mL; supernatants were subsequently supplemented [55].

## Plus-strand RNA virus replication model

We developed a mechanistic model using ordinary differential equations (ODEs) and mass action kinetics to analyze pan-viral similarities and virus-specific differences within the plus-strand RNA virus life cycle. Our published models on two plus-strand RNA viruses, HCV and DENV, served as a basis for the pan-viral plus-strand RNA virus replication model [19,55,57]. However, in our previous published models, we studied host dependency factors responsible for cell line permissiveness and restriction factors such as the innate immune response. Therefore, those models were modified to reflect merely the plus-strand RNA life cycle from virus entry to release of all viruses considered here.

The resulting model of plus-strand RNA virus replication is composed of four main processes: Entry of plus-strand RNA virus via receptor-mediated endocytosis and release of the viral genome (Fig 1 steps ① and ②), its subsequent translation into viral proteins (Fig 1 steps ③ to ⑤), viral RNA replication within the replication organelle (Fig 1 steps ⑥ to ⑨), and further replication (Fig 1 step ⑩) or RNA export out of the replication organelle (Fig 1 step ⑪) or virus packaging and release from the cell with subsequent reinfection of the same cell or infection of naïve cells (Fig 1 steps ⑫ and ⑬).

The virus infection process (Eqs 1 and 2), i.e., receptor-mediated virus entry, fusion, and release of the viral genome into the cytoplasm, as well as reinfection of the same cell or further infection of naïve cells (Eq 14) are represented by extracellular virus $V$, virus within endosomes $V_E$, and newly produced virus released from infected cells $V_R$ and are given by the equations

$$\frac{dV}{dt} = -k_e^i V + k_{re} V_R - \mu_V^i V \tag{1}$$

and

$$\frac{dV_E}{dt} = k_e^i V - k_f^i V_E - \mu_{V_E} V_E. \tag{2}$$

Extracellular virus $V$ enters a single cell via receptor-mediated endocytosis with rate constant $k_e^i$ or degrades with constant rate $\mu_V^i$. Note that virus-specific parameters are marked with superscripted $i$ with $i \in \{HCV, DENV, CVB3\}$. The virus within endosomes $V_E$ either degrades with rate constant $\mu_{VE}$ or undergoes conformational changes of its nucleocapsid resulting in the release of the viral genome $R_P$ with rate constant $k_f^i$. Note that extracellular virus is also replenished by the release of virus from the cell at rate $k_{re}$.

Viral RNA translation and replication (Eqs 3 to 13) are modeled based on our published HCV and DENV models [19,55]. In brief, our model describes the translation-associated processes in the cytoplasm (Eqs 3 to 8) starting with free viral RNA $R_P$ in the cytoplasm, an intermediate translation initiation complex $TC$, as well as the translated polyprotein $P_P$ which is cleaved into structural and non-structural viral proteins, $P_S$ and $P_N$, respectively. Note that a firefly luciferase gene has been integrated into the viral genomes. The luciferase activity $L$ was measured from cell lysates as a marker for translation activity (see Methods) reflecting protein concentration and has been introduced into the model. Translation and polyprotein processing are modeled with the following ODEs, where $Ribo_{tot}^i$ and $RC_{MAX}$ are the total number of

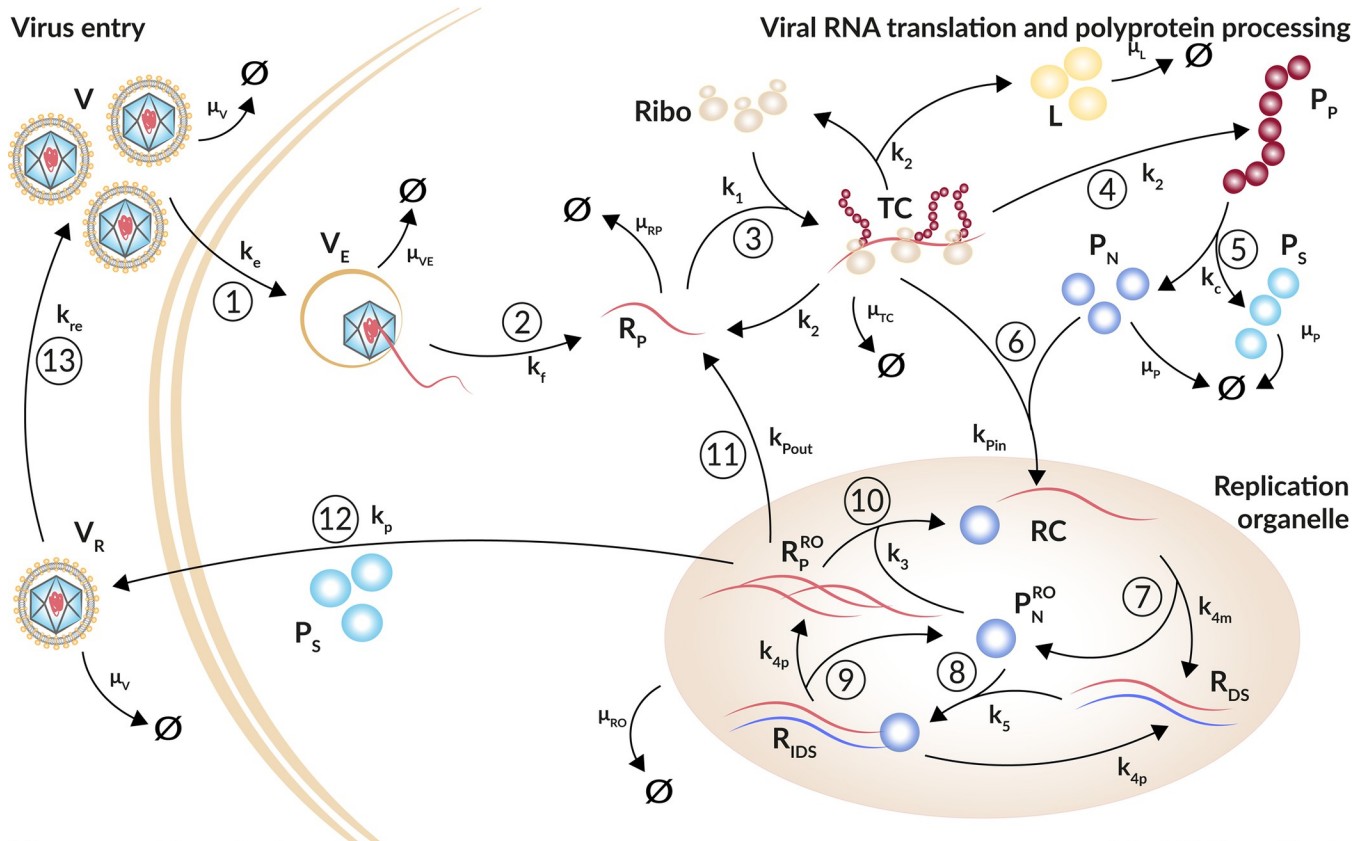

**Fig 1. Schematic illustration of the plus-strand RNA life cycle.** ① Virus ($V$) enters the cell via receptor-mediated endocytosis ($k_e$). ② The viral genome ($R_P$) is released ($k_f$). Virus within the endosome ($V_E$) degrades with rate constant $\mu_{VE}$. ③ Ribosomes ($Ribo$) bind at the viral genome and form ($k_1$) a translation initiation complex ($TC$) that degrades with rate constant $\mu_{TC}$. ④ The viral genome ($R_P$) is translated ($k_2$) into a polyprotein ($P_P$) that ⑤ is subsequently cleaved ($k_c$) into structural and non-structural viral proteins, $P_S$ and $P_N$, respectively. To measure translation activity, luciferase ($L$) is integrated into the viral genome and produced with RNA translation. Viral proteins degrade with rate constant $\mu_P$; luciferase degrades with rate constant $\mu_L$. ⑥ Non-structural proteins and freshly translated viral RNA form ($k_{Pin}$) replicase complexes ($RC$) that are associated with replication organelles (ROs) and ⑦ serve as a template for the minus-strand synthesis ($k_{4m}$) leading to double-stranded RNA ($R_{DS}$). ⑧ Viral non-structural proteins, such as the RdRp, within the replication organelle ($P_N^{RO}$) bind to double-stranded RNA, forming ($k_5$) a minus-strand replication intermediate complex ($R_{IDS}$) that ⑨ initiates the plus-strand RNA synthesis ($k_{4p}$) giving rise to multiple copies of viral plus-strand RNA ($R_P^{RO}$). All species within the replication organelle degrade with the same rate constant $\mu_{RO}$. ⑩ The viral genome can remain within the replication organelle, where it undergoes multiple rounds of genome replication ($k_3$), ⑪ it can be exported ($k_{Pout}$) out of the replication organelle into the cytoplasm starting with the translation cycle again, or ⑫ the plus-strand RNA genome ($R_P^{RO}$) is packaged together with structural proteins ($P_S$) into virions ($V_R$) that are released from the cell ($k_p$) and ⑬ may re-infect the same cell or infect naïve cells ($k_{re}$). Extracellular infectious viral species ($V$ and $V_R$) degrade with rate constant $\mu_V$.

ribosomes and maximal number of replicase complexes in a cell (see below for details), respectively:

$$\frac{dR_P}{dt} = k_f^i V_E - k_1 R_P \left( Ribo_{tot}^i - TC \right) + k_2^i TC + k_{Pout}^i R_P^{RO} - \mu_{RP}^i R_P, \tag{3}$$

$$\frac{dTC}{dt} = k_1 R_P \left( Ribo_{tot}^i - TC \right) - k_2^i TC - k_{Pin}^i \left( 1 - \frac{RC}{RC_{MAX}} \right) P_N TC - \mu_{TC}^i TC, \tag{4}$$

$$\frac{dP_P}{dt} = k_2^i TC - k_c P_P, \tag{5}$$

$$\frac{dL}{dt} = k_2^i TC - \mu_L L, \tag{6}$$

$$\frac{dP_S}{dt} = k_c P_P - \mu_P^i P_s - N_{P_S}^i \nu_p, \tag{7}$$

$$\frac{dP_N}{dt} = k_c P_P - k_{Pin}^i \left(1 - \frac{RC}{RC_{MAX}}\right) P_N TC - \mu_P^i P_N. \tag{8}$$

With rate constant $k_1$, free host ribosomes form a translation complex $TC$ with the viral plus-strand RNA genome $R_P$. The total number of ribosomes ($Ribo_{tot}^i$) available for viral RNA translation was assumed constant, and $Ribo = Ribo_{tot}^i - TC$ gives the number of free ribosomes. Note that $Ribo_{tot}^i$ is only a fraction of the total cellular ribosome number. Translation of the viral plus-strand RNA genome generates the viral polyprotein $P_P$ and luciferase $L$ with rate constant $k_2^i$. The viral polyprotein $P_P$ is subsequently cleaved with rate constant $k_c$ into structural and non-structural viral proteins, $P_S$ and $P_N$, respectively. The translation complex $TC$ decays with rate constant $\mu_{TC}^i$, while luciferase and viral proteins degrade with rate constants $\mu_L$ and $\mu_P^i$, respectively. Note that for simplicity, we assume structural and non-structural proteins degrade with the same rate constant, which has been summarized as one virus-specific viral protein degradation rate $\mu_P^i$.

The subsequent processes of viral RNA synthesis in the replication organelle (RO) are modeled by Eqs 9 to 13, representing the replicase complex $RC$, double-stranded RNA $R_{DS}$, a double-stranded RNA intermediate complex $R_{IDS}$, newly synthesized viral plus-strand RNA in the RO $R_P^{RO}$, and non-structural proteins within the RO $P_N^{RO}$, as follows:

$$\frac{dRC}{dt} = k_{Pin}^i (1 - \frac{RC}{RC_{MAX}}) P_N TC - k_{4m}^i RC + k_3 R_P^{RO} P_N^{RO} - \mu_{RO} RC, \tag{9}$$

$$\frac{dR_{DS}}{dt} = k_{4m}^i RC - k_5 R_{DS} P_N^{RO} + k_{4p}^i R_{IDS} - \mu_{RO} R_{DS}, \tag{10}$$

$$\frac{dR_{IDS}}{dt} = k_5 R_{DS} P_N^{RO} - k_{4p}^i R_{IDS} - \mu_{RO} R_{IDS}, \tag{11}$$

$$\frac{dP_N^{RO}}{dt} = k_{4m}^i RC - k_3 R_P^{RO} P_N^{RO} - k_5 R_{DS} P_N^{RO} + k_{4p}^i R_{IDS} - \mu_{RO} P_N^{RO}, \tag{12}$$

$$\frac{dR_P^{RO}}{dt} = k_{4p}^i R_{IDS} - k_3 R_P^{RO} P_N^{RO} - k_{Pout}^i R_P^{RO} - \nu_p - \mu_{RO} R_P^{RO}. \tag{13}$$

Viral non-structural proteins recruit the viral RNA after translation to the replicase complex [58]. Hence, for viral RNA synthesis, we require translated viral RNA, i.e., the translation complex $TC$ instead of free cytosolic viral RNA $R_P$, to interact with the non-structural proteins. Thus, the translation complex $TC$ and a subset of non-structural proteins $P_N$ are imported into the RO, leading to the formation of a replicase complex $RC$ with rate constant $k_{Pin}^i$. Following successful replicase complex formation, ribosomes dissociate from the complex as is accounted for in Eq (4). We furthermore assume that there is a limitation in the number of replicase

complexes formed within a cell. To do so, we extend $k_{Pin}^i$ by $\left(1 - \frac{RC}{RC_{MAX}}\right)$ with the carrying capacity for replicase complexes $RC_{MAX}$ [57,59].

Within the RO, minus-strand RNA synthesis occurs from the replicase complex with rate constant $k_{4m}^i$, leading to the formation of double-stranded RNA $R_{DS}$, which along with the non-structural proteins, are released from the RC, $P_N^{RO}$. Subsequently, the double-stranded RNA binds again to $P_N^{RO}$ with rate constant $k_5$ to form a double-stranded intermediate replicase complex $R_{IDS}$, initiating plus-strand RNA synthesis with rate constant $k_{4p}^i$. For simplicity, we assume that minus and plus-strand RNA synthesis occur with the same rate constant $k_{4m}^i = k_{4p}^i$. The newly synthesized plus-strand RNA genomes $R_P^{RO}$ either remain within the RO to make additional replicase complexes with rate constant $k_3$, are exported out of the RO into the cytoplasm for further RNA translation with export rate $k_{Pout}^i$, or are packaged together with structural proteins into virions $V_R$ and are subsequently released from the cell. Assembly and release of virus particles is represented by a Michaelis-Menten type function $v_p$ described below (Eq 15, compare [55,60]). The RNA and protein species within the RO ($RC$, $R_{DS}$, $R_{IDS}$, $R_P^{RO}$, $P_N^{RO}$) are assumed to degrade with the same decay rate $\mu_{RO}$ and represent the decay of the entire replication organelle.

The released virus $V_R$ may re-infect the same cell or infect new cells with rate constant $k_{re}$, or degrade with rate constant $\mu_V^i$, resulting in the equation

$$\frac{dV_R}{dt} = v_p - k_{re}V_R - \mu_V^i V_R. \tag{14}$$

Assembly of newly synthesized viral plus-strand RNA genome $R_P^{RO}$ and viral structural proteins $P_S$ into viral particles and their subsequent release from the host cell are described using a Michaelis-Menten type function, with rate

$$v_p = k_p R_P^{RO} \frac{P_S}{K_D^i N_{P_S}^i + P_S}, \tag{15}$$

where $k_p$ is the virion assembly and release rate, and $k_p R_P^{RO}$ is the maximum release rate limited by viral resources. Let $N_{P_S}^i$ be the number of structural proteins in a virus of type $i$, then to produce virus at rate $v_p$ will require a large number of proteins $K_D^i N_{P_S}^i$, where $K_D^i$ is a scaling constant and $K_D^i N_{P_S}^i$ is the number that corresponds to the half-maximal release rate [see [55,60,61] for more details].

## Pan-viral and virus-specific model parameters

To complete the plus-strand RNA virus model, we need to specify model parameters. To prevent overfitting and parameter uncertainty, we fixed many parameter values to either experimentally determined values or values estimated in other modeling studies. In some cases, we could calculate velocities directly, such as for viral RNA translation and synthesis, which could thus be fixed as described in S1 Text. An overview of all parameter values is given in Table 2.

## Parameter estimation, model selection, and model analysis

Our model has 61 parameters; 30 of them were fixed, while 31 were estimated by fitting the model to the experimental data. As the fixed parameter values were experimentally measured, calculated, or taken from literature, we had information about which were virus-specific (S1 Text and Table 2). To determine which of the remaining model parameters are conserved across the different viruses considered (pan-viral) and which parameters are virus-specific, we

**Table 2. Parameter values and 95% confidence intervals in ().** Note that parameter values marked with * were fixed due to previous assumptions and calculations. Furthermore, confidence intervals marked with + hit the set estimation boundary; ± calculated from the data; # experimentally measured for Zika virus; ‡ experimentally measured for poliovirus.

| Parameter | Description | HCV | DENV | CVB3 | Unit |
|---|---|---|---|---|---|
| $k_e^i$ | Virus entry rate | 10 (1.9, $10^+$) | 0.31 (0.28, 0.34) | 1.3 (0.9, 1.7) | 1/h |
| $k_f^i$ | RNA release rate | 10 (1.7, $10^+$) | 0.008 (0.006, 0.01) | 0.016 (0.006, 0.04) | 1/h |
| $k_1$ | Formation rate of the translation complex | 1000 (840, $1000^+$) | | | mL/molecule /h |
| $k_2^i$ | Virus RNA translation rate | 180 [65] | 100 [55] | 300 ‡ [66] | 1/h |
| $k_c$ | Polyprotein cleavage rate | 2.24 (1.18, 7.4) | | | 1/h |
| $k_3$ | Formation of additional replicase complexes within the replication organelle | 42 (5.5, 525) | | | mL/molecule /h |
| $k_{4m}^i = k_{4p}^i$ | Minus- and plus-strand RNA synthesis rate | 1.1 [65] | 1.0 [55] | 50 ‡ [66] | 1/h |
| $k_{Pin}^i$ | Formation rate of the replicase complex | 4.4 (2.4, 7.5) | 0.45 (0.29, 0.74) | 1.4 (0.52, 4.09) | mL/molecule /h |
| $k_5$ | Formation rate of the replication intermediate complex | 6018 (1549, 68401) | | | mL/molecule /h |
| $k_{Pout}^i$ | Export rate of viral RNA out of the replication organelle | 33 (0.8, 1477) | 53 (16, 432) | 0.23 (0.16, 0.43) | 1/h |
| $k_p$ | Assembly and release rate | 158 (47, $1000^+$) | | | mL/molecule /h |
| $k_{re}$ | Reinfection rate | 0.01 ($0.01^+$, 0.038) | | | 1/h |
| $\mu_{RP}^i$ | Degradation rate of cytosolic viral RNA | 0.26 [65] | 0.23 [67] | 0.15 ‡ [68] | 1/h |
| $\mu_{TC}^i$ | Degradation rate of the translation complex | 0.13 * | 0.115 * | 0.075 * | 1/h |
| $\mu_{RO}$ | Degradation rate of viral RNA and protein within the replication organelle | 0.0842 [19] | | | 1/h |
| $\mu_P^i$ | Degradation rate of viral protein | 0.08 [19] | 0.46 [67] | 0.43 [69] | 1/h |
| $\mu_L$ | Degradation rate of luciferase | 0.35 [19] | | | 1/h |
| $\mu_V^i$ | Degradation rate of extracellular infectious virus | 0.1 [57] | 0.13 [70] | 0.08 [71,72] | 1/h |
| $\mu_{VE}$ | Degradation rate of intracellular virus within the endosome | 0.23 # [73] | | | 1/h |
| $V_0^i$ | Initial virus concentration | 0.2 (0.16, 0.25) | 1 (0.8, 1.3) | 1 (0.4, 2.2) | molecules/mL |
| $Ribo_{tot}^i$ | Total ribosome concentration | 0.005 (0.004, 0.007) | 0.48 (0.41, 0.55) | 6.7 (5.0, 9.1) | molecules |
| $RC_{MAX}$ | Maximum number of replicase complexes | 0.46 (0.34, 0.64) | | | molecules/mL |
| $K_D^i$ | Scaling constant for virus | 0.04 ± | 1.8 ± | 40 ± | virions |
| $N_{P_S}^i$ | Number of structural proteins needed to produce 1 virion | 180 [65,74] | 180 [55,74] | 60 [15] | molecules/ virion |
| $f_{R_P}^i$ | Scale factor for plus-strand RNA | 394 (274, 524) | 0.76 (0.58, 1.0) | 550 (245,1366) | |
| $f_{R_M}^i$ | Scale factor for minus-strand RNA | 1377 (945, 1872) | - | - | |
| $f_L^i$ | Scale factor for luciferase | - | 0.41 (0.33, 0.5) | 0.08 (0.06, 0.1) | |

performed several rounds of model evaluation using the Akaike information criterion (AIC) and model identifiability analysis (profile likelihood estimation). See S2 Text for a description of the model selection process.

We simultaneously fit the plus-strand RNA virus replication model to the virus-specific data sets for HCV, DENV, and CVB3. To fit the mathematical model to the experimental data, we calculated the total plus-strand RNA $R_P^{tot} = (V_E + R_P + TC + RC + R_{DS} + R_{IDS} + R_P^{RO})$, total minus-strand RNA $R_M^{tot} = (R_{DS} + R_{IDS})$, luciferase $L$, and total infectious virus $V^{tot} = (V + V_R)$. Note that our model accounts for infectious virus since infectious titers were measured for all three viruses. Further note that for the infectious virus measurements for HCV, $V^{tot} = V_R$, since measuring infectious virus started 20 h pi. We introduced three scale factors $f_L, f_{R_M}$, and $f_{R_P}$ to re-scale experimental measurements acquired in relative measurements (plus-strand

RNA for DENV), molecules per cell (plus- and minus-strand RNA measurements for HCV and plus-strand RNA for CVB3), and relative light unit (luciferase for DENV and CVB3).

We implemented the model in MATLAB (The MathWorks) 2016 using the Data2Dynamics toolbox [62]. We assessed model identifiability using the profile likelihood estimation method implemented in Data2Dynamics [62,63]. In Data2Dynamics, a parameter is identifiable if its 95% confidence interval is finite [62,63]. Note that an estimated model parameter may hit a predefined upper or lower parameter boundary which hampers the calculation of the 95% confidence interval. In such cases, a one-sided 95% confidence interval has been calculated starting from the estimated model parameter and thus with its upper or lower boundary marked with + in Table 2. Details about the model fitting and model selection process are in S2 Text.

We performed a global sensitivity analysis in MATLAB using the extended Fourier Amplitude Sensitivity Test (eFAST) [64]. We calculated sensitivities with regard to the total plus-strand RNA ($R_p^{tot}$) concentrations throughout the course of infection. We studied hypothetical drug interventions by including the effects of direct-acting antivirals (DAA) into the model. For this purpose, we simulated putative drugs targeting (1) viral entry and internalization $k_e$, (2) release of the viral RNA genome $k_f$, (3) formation of the translation initiation complex $k_1$, (4) viral RNA translation $k_2$, (5) polyprotein cleavage $k_c$, (6) replicase complex formation $k_{Pin}$, (7) minus- and plus-RNA synthesis $k_{4m}$ and $k_{4p}$, as well as (8) virus particle production and release ($v_p$). To introduce drug effects into the model, we assumed a drug efficacy parameter $0 \leq \varepsilon \leq 1$, and multiplied the parameters above by $(1-\varepsilon)$ to simulate drug treatment. Similar to our previously published DENV model, we calculated the average virus particle concentration released from the cell upon drug administration ($\varepsilon \neq 0$) until 5 days post-drug administration, i.e., a drug treatment observation window of 120 h. The average virus particle concentration with treatment ($\varepsilon \neq 0$) has been normalized to the average virus concentration without drug treatment ($\varepsilon = 0$). Note that we studied two different time points of drug administration: at the very beginning of the infection, 0 h pi, and when the system is in steady state, 100 h pi.

## Results

As shown on the left in Fig 2, the model replicates the experimental data for all three viruses. Virus-specific characteristics are revealed by comparing the dynamics of the three viruses and their plus-strand RNA genomes. CVB3 is a fast-replicating virus with a life cycle duration of about 8 hours (depending on the cell type), after which the infected cells begin to die. Similarly, DENV is cytopathic but seems to be slower replicating and thus has a longer life cycle than CVB3, with DENV starting to produce virus about 16 h pi [56]. In contrast, HCV is non-cytopathic with a consequently longer life cycle. In our experimental measurements, the CVB3 viral load peaked at the end of its life cycle with 193 PFU/mL/cell. The HCV viral load peaked at 0.06 PFU/mL/cell around 44 h pi, while the DENV viral load reached its maximum with approximately 8 PFU/mL/cell around 10 hours earlier at 30 to 34 h pi (Fig 2A, 2B, and 2C). We calculated the corresponding average virus concentration per measurement time point for HCV, DENV, and CVB3 per cell as 0.04 PFU/mL/cell, 1.8 PFU/mL/cell, and 40 PFU/mL/cell, respectively. Thus, the average infectious HCV viral load was only 4% of the average DENV viral load and only 0.3% of the average CVB3 viral load. Similarly, CVB3 reached a peak of almost 500,000 plus-strand RNA copies per cell at 8 h pi, while HCV produced only 10,000 copies per cell at 70 h pi, i.e., 98% less than CVB3. Note that both, CVB3 infectious virus and plus-strand RNA, increased several 1000-fold in time.

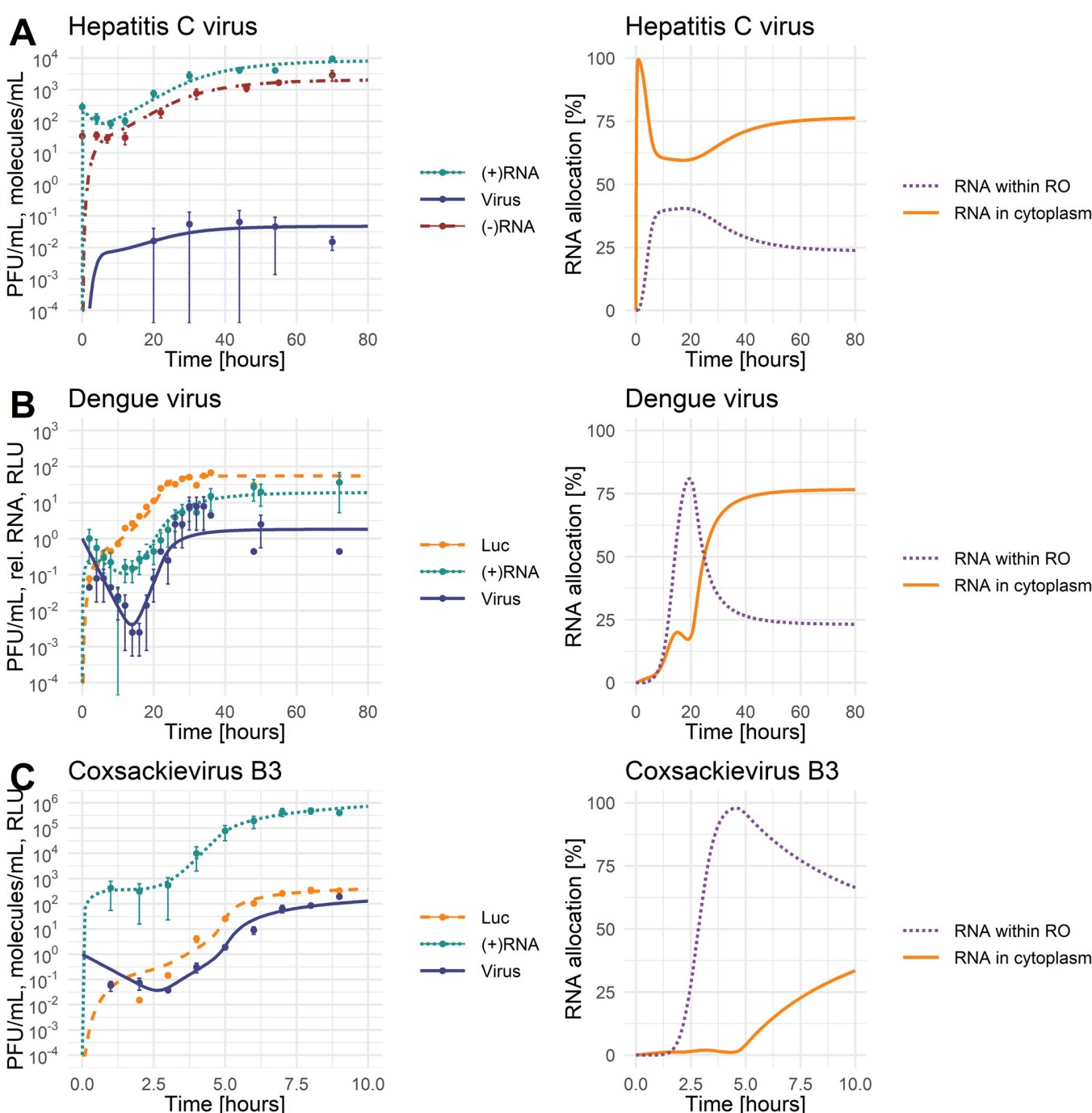

**Fig 2.** Best fit of the model to the data with standard deviation (left panel) and model prediction of plus-strand RNA allocation between the cytoplasm and replication organelle (RO) (right panel). For parameter values, see Table 2. [LEFT: green: (+)RNA = $R_P^{tot} = (V_E + R_P + TC + RC + R_{DS} + R_{IDS} + R_P^{RO})$, red: (-) RNA = $R_M^{tot} = (R_{DS} + R_{IDS})$, blue: A) Virus = $V^{tot} = V_R$, B) and C) Virus = $V^{tot} = (V + V_R)$, yellow: Luc = L; RIGHT: yellow: RNA in cytoplasm = $(R_P + TC)/R_P^{tot}$, purple: RNA within replication organelle (RO) = $RC + R_{DS} + R_{IDS} + R_P^{RO})/R_P^{tot}$; Infectious virus was measured in PFU/mL, (+) and (-)RNA were measured in molecules/mL or relative RNA concentration, luciferase was measured in relative light unit (RLU)].

## Model selection and uncertainty

The intracellular model structure has been taken from our previously published HCV model [19], upon which we built with our recently published DENV model [55]. However, a striking difference from our previous HCV and DENV models is the absence of host factors involved in replicase complex formation and virus assembly and release. We have previously shown that host factors are recruited by the virus and seem beneficial for host cell permissiveness and virus replication efficiency [19,55]. Instead, here we describe intra-viral replication differences with virus-specific parameter sets based on model evaluation by AIC and profile likelihood estimation (see Methods, S1 and S2 Texts).

Considering the maximal number of replicase complexes ($RC_{MAX}$) improved the basic model AIC from 3025 to 1982 and thus served as a starting point for the virus-specific model selection process (see S2 Text). After several rounds of model selection by comparing AICs and taking model identifiability into account, we added five virus-specific processes into our basic model (from a total of 13 considered processes): (1) the total number of ribosomes $Ribo^i_{tot}$ available for viral RNA translation, (2) virus entry $k^i_e$, (3) viral genome release $k^i_f$, (4) formation of the replicase complex $k^i_{Pin}$, and (5) export of viral RNA from the RO into the cytoplasm $k^i_{Pout}$. Note that based on literature data and previous assumptions, we fixed some virus-specific and pan-viral processes and degradation rates (see S1 Text and Table 2). The best-fit model showed high similarity to the virus-specific experimental measurements and a high degree of model identifiability (see Fig 2 for best fit, Fig 3 for the parameter profiles based on the profile likelihood estimation, and Table 2 for parameter values with 95% confidence intervals).

## RNA allocation

As predicted by our model, the allocation of plus-strand RNA in the cytoplasm and within the RO shows interesting virus-specific differences (Fig 2 right panel). Compared to the total amount of viral RNA, HCV has most of the RNA allocated to the cytoplasm and is thus available for viral RNA translation at any given time. In DENV, our model predicted that the allocation strategy changes throughout the viral life cycle, with most plus-strand RNA within the RO initially. At around 25 h pi, viral RNAs are equally distributed between the two compartments, while at the end of the DENV life cycle, the majority of viral RNA is in the cytoplasm. Interestingly, in steady state, the predicted allocation of both HCV and DENV is the same, with 25% of RNA allocated to the RO and 75% to the cytoplasm. In contrast, the predicted viral RNA allocation is the opposite for CVB3. CVB3 has the majority of RNA available within the RO, contributing to the 2- to 3-log higher viral load.

## Virus-specificity

For a successful virus infection, the first hurdles to overcome are virus entry and the release of the viral genome into the cytoplasm. The rate constants for virus entry $k^i_e$ and vRNA release $k^i_f$ had the highest estimated values for HCV. However, both values were practically non-identifiable, suggesting a limitation in the amount of data. Hence, we could only estimate the lower boundary of the 95% confidence intervals, which suggests $k^{HCV}_e \geq 1.9\ h^{-1}$ and $k^{HCV}_f \geq 1.7\ h^{-1}$. CVB3 seems slightly better adapted to the cell line with a 4-times higher entry rate and 2-times higher vRNA release rates than DENV. According to our model selection process, the degradation rate of the internalized virus within endosomes $\mu_{VE}$ was pan-viral, suggesting neither an advantage nor disadvantage for the studied viruses.

## Parameter identifiability profile

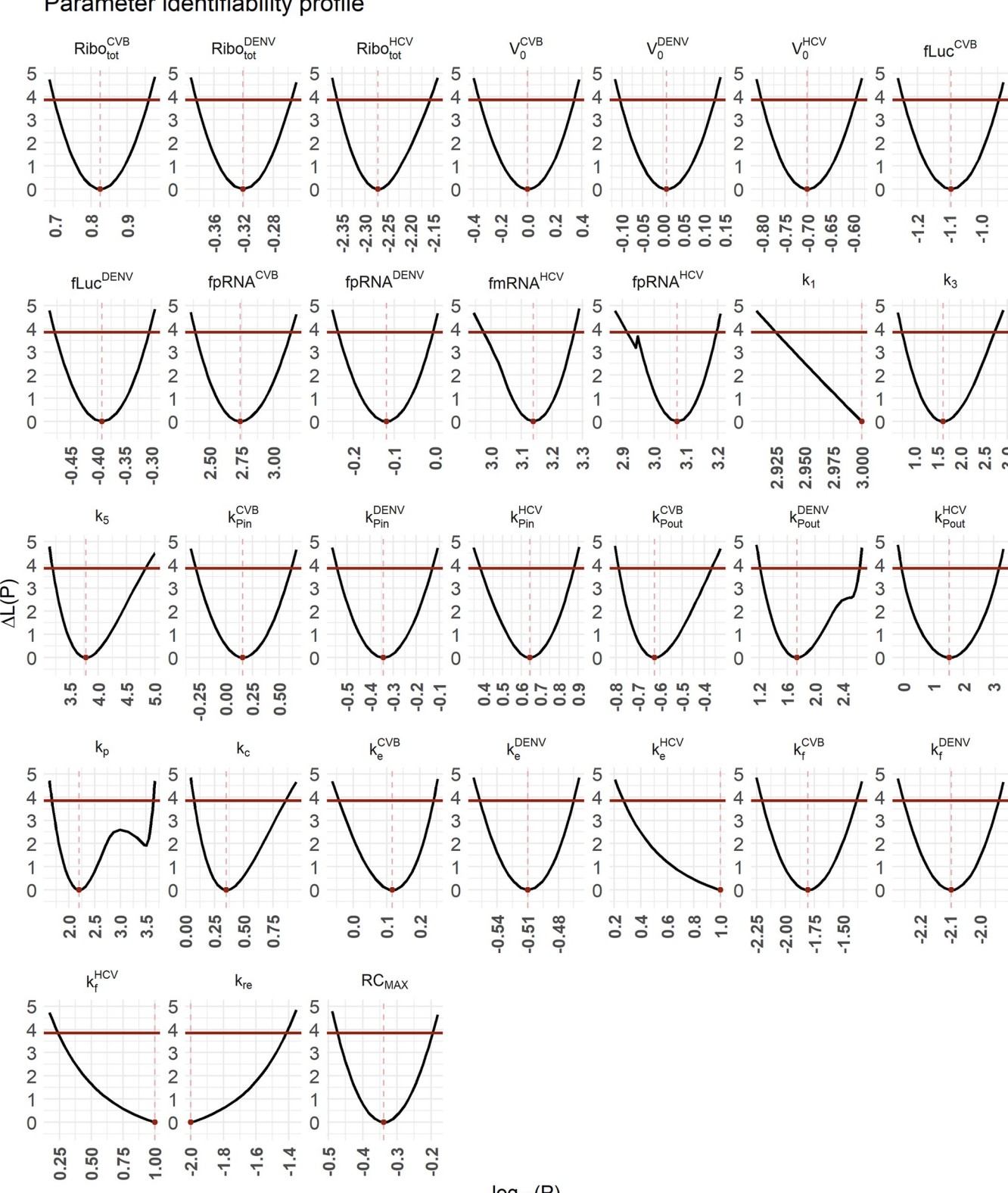

**Fig 3. Uncertainty analysis of the best-fit model.** For parameter values and 95% confidence intervals, see Table 2. The best fit is shown in Fig 2.

The following processes in the viral life cycle are vRNA translation and polyprotein processing with parameters $k_1$ for the formation of the translation initiation complex, $k_2^i$ vRNA translation, and $k_c$ polyprotein cleavage. Models including virus-specific $k_1$ or $k_c$ either did not improve the quality of the model fit (no AIC improvement) or were non-identifiable when tested as virus-specific and thus have been selected as pan-viral (see S2 Text). However, the viral RNA translation rate $k_2^i$ was calculated based on genome size and ribosome density and set as virus-specific (see S1 Text). In the vRNA translation and polyprotein processing step, our model selected the total number of ribosomes $Ribo_{tot}^i$ as the only virus-specific parameter. Since the ribosome number has been selected in the first round of model selection (see S2 Text), it emphasizes the importance of this host factor, with CVB3 showing the highest estimated ribosome number available for RNA translation. In contrast, HCV and DENV use only 0.07% and 7% of the ribosomes CVB3 uses, respectively. Interestingly, increasing the number of ribosomes in the HCV life cycle to those of CVB3 (from $Ribo_{tot}^{HCV} = 0.005$ to $Ribo_{tot}^{HCV} = 6.7$ molecules per mL) increases the infectious virus load by three orders of magnitude (Fig 4A). In the same way, decreasing the number of ribosomes in the CVB3 life cycle to those of HCV (from $Ribo_{tot}^{CVB3} = 6.7$ to $Ribo_{tot}^{CVB3} = 0.005$ molecules per mL) decreases the CVB3 virus load by three orders of magnitude (Fig 4B). In contrast, when increasing the viral RNA synthesis rates of HCV to those of CVB3 (from $k_{4m}^{HCV} = k_{4p}^{HCV} = 1.1$ to $k_{4m}^{HCV} = k_{4p}^{HCV} = 50 \ h^{-1}$), the viral load did not increase. However, decreasing the viral RNA synthesis rates of CVB3 to those of HCV (from $k_{4m}^{CVB3} = k_{4p}^{CVB3} = 50$ to $k_{4m}^{CVB3} = k_{4p}^{CVB3} = 1.1 \ h^{-1}$) decreased the viral load by one order of magnitude. This suggests an important role of ribosomes as key players in the production of structural and non-structural proteins necessary for efficient vRNA replication and virus production.

The subsequent processes of the vRNA replication depend on successful viral protein production. Viral non-structural proteins are crucial for forming the replicase complex and its formation rate $k_{Pin}^i$, which has been selected as virus-specific. Here, HCV seems to be more efficient and better adapted to the Huh7 cell line, showing a 10- and 4-times faster formation rate compared to DENV and CVB3, respectively. Furthermore, our estimated replicase complex formation rates suggest that the formation of double-membrane vesicles may be more efficient (HCV and CVB3) compared to the formation of invaginations (DENV). However, the maximum number of replicase complexes $RC_{MAX}$ and the degradation of species within the RO ($\mu_{RO}$) were not selected as virus-specific, especially since the viral RNA synthesis rates were initially set as virus-specific (Table 2). Interestingly, even though being a pan-viral model parameter, not all viruses reached the maximal number of replicase complexes $RC_{MAX}$ (the carrying capacity). The dynamics of replicase complexes show a clear separation between DENV and CVB3 versus HCV (Fig 5A and 5B). CVB3 reached the estimated carrying capacity of around 5 h pi, while DENV reached 98% of the possible carrying capacity of around 25 h pi. Strikingly, the replicase complex formation for HCV reached its maximum at a 74% lower level of the pan viral carrying capacity, even though our model estimated the fastest RC formation rate for HCV.

The export of viral RNA from the RO to the site of RNA translation $k_{Pout}^i$ has also been selected as virus-specific, where HCV and DENV seem more efficient than CVB3, which showed an almost 190 times slower trafficking process.

Following the production of viral proteins and RNA genomes, the single components assemble into virions and are released from the cell. Here, the virus assembly and release rate $k_p$ and the reinfection rate $k_{re}$ have been selected as pan-viral. Note that the scaling constant $K_D^i$ and the number of structural proteins necessary per virion $N_{P_S}^i$, were calculated from the data or taken from the literature, respectively, and thus set as virus-specific (Table 2).

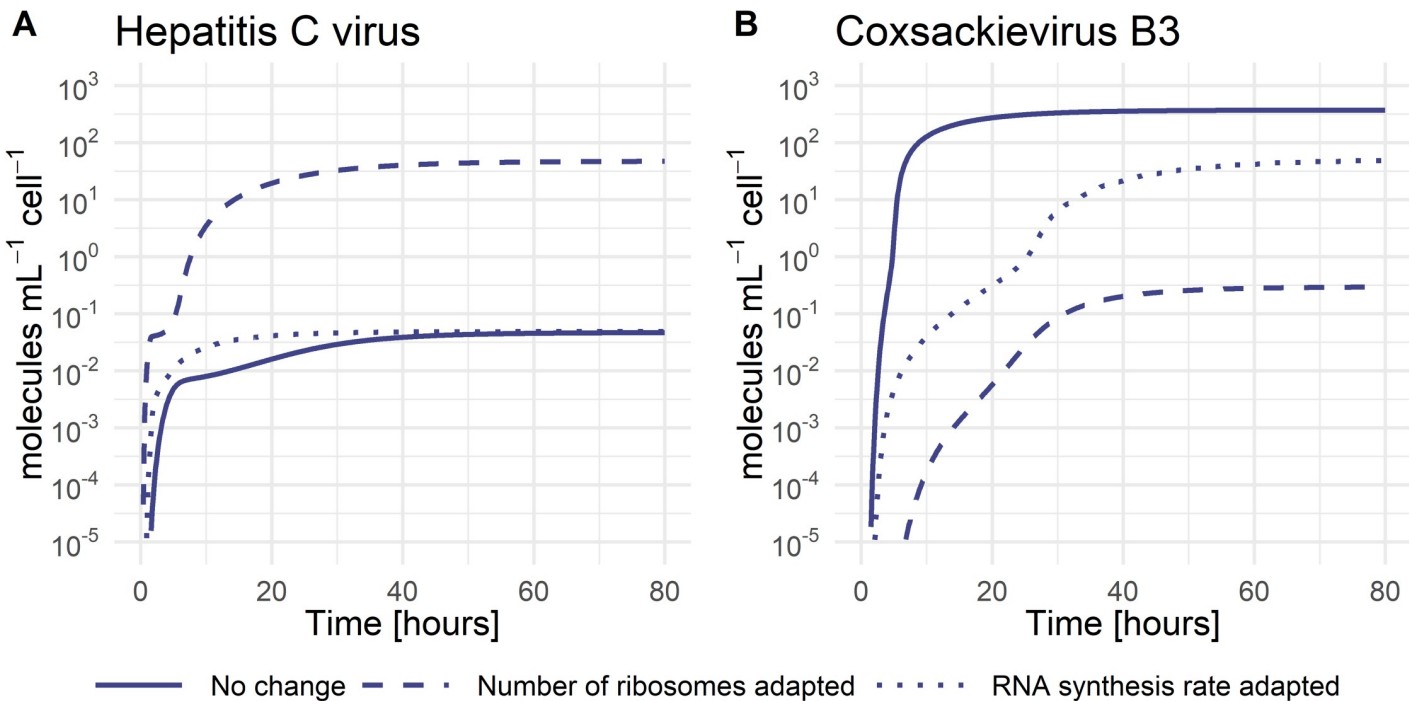

**Fig 4. Infectious virus concentration with parameter adjustments. A)** HCV concentration with estimated parameters (solid), the number of ribosomes taken from CVB3 (dashed), and the RNA synthesis rate taken from CVB3 (dotted). **B)** CVB3 concentration with estimated parameters (solid), the number of ribosomes taken from HCV (dashed), and the RNA synthesis rate taken from HCV (dotted).

## Sensitivity analysis and drug intervention

Having a detailed model of the intracellular replication of plus-strand RNA viruses, we next addressed the question of which processes shared across all viruses showed the highest sensitivity index to potential drug interventions (Fig 6). Our sensitivity analysis suggests that model parameters associated with vRNA translation ($k_2^i$) and synthesis within the RO ($k_{4m}^i$ and $k_{4p}^i$) are highly sensitive for all viruses. Furthermore, all viruses were sensitive to the formation of replicase complexes $k_{Pin}^i$ and its maximum number $RC_{MAX}$.

Interestingly, DENV and CVB3 showed a time-dependent sensitivity pattern over the course of infection, beginning with viral entry ($k_e^i$) being sensitive, followed by the release of the viral genome ($k_f^i$). However, both model parameters were not sensitive to HCV, possibly due to practical non-identifiability (see above). Moreover, vRNA translation and replication seem to start around 5 or 20 h pi in CVB3 and DENV, respectively, suggesting viral entry as a rate-limiting process.

There are also some interesting differences between the three viruses. While the formation of the translation initiation complex ($k_1$) showed a higher sensitivity in HCV, vRNA translation ($k_2^i$) was more sensitive for CVB3 and DENV. Furthermore, for HCV, the number of ribosomes available for HCV RNA translation was one of the most sensitive parameters while having negligible sensitivity for CVB3 and DENV. This may reflect the strength of the internal ribosome entry site, IRES, (CVB3) or the 5' UTR/Cap (for DENV), where a strong IRES may require fewer ribosomes for robust recruitment to initiate vRNA translation. However, for CVB3, viral RNA export $k_{Pout}^i$ is among the most sensitive processes, while being not sensitive for HCV and DENV. Interestingly, the degradation of virus in endosomes ($\mu_{VE}$) showed the highest sensitivity among the degradation rates for DENV early in infection (around 10 to 25

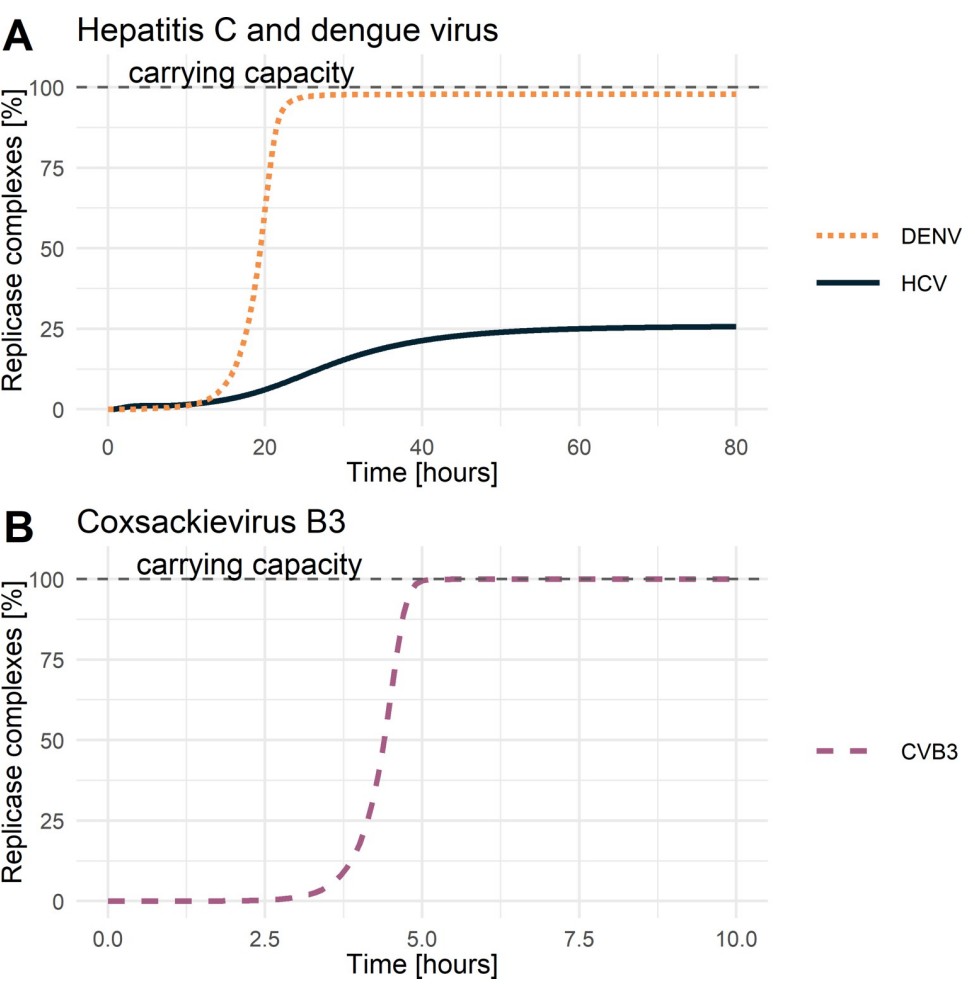

**Fig 5. Replicase complexes over time.** Dynamics of replicase complexes for **A)** hepatitis C and dengue virus, **B)** coxsackievirus B3. The dashed grey line represents the carrying capacity or the maximum number of formed replicase complexes.

h pi). In contrast, the degradation of cytosolic vRNA ($\mu_{RP}$) seems highly sensitive towards the end of infection for both DENV and CVB3.

As a next step, we aimed to analyze if any processes can be targeted, leading to a 99% reduction in extracellular virus upon inhibition. We, therefore, studied the effects of inhibiting core processes of the viral life cycle (Fig 7). We then simulated *in silico* the administration of a hypothetical drug at two different time points using our mathematical model: at the beginning of the infection (0 h pi) or in steady state (100 h pi). The drug administration at the beginning of infection (0 h pi) will give insights into infection prevention. The drug administration in a steady state (100 h pi) has the advantage of studying the system in the equilibrium of vRNA replication/virus production and vRNA degradation/virus clearance and, thus, how to treat an established infection. Therefore, we can ignore a potential bias of the drug effect when the vRNA translation and replication machinery must be established or host cellular and viral resources are exhausted, leading to inefficient viral RNA replication and, ultimately, virus production. Even though DENV and CVB3 are viruses that cause acute infections, cleared after a couple of weeks, studying both viruses in a steady state is important to gain insights about a possible drug effect during maintained virus production.

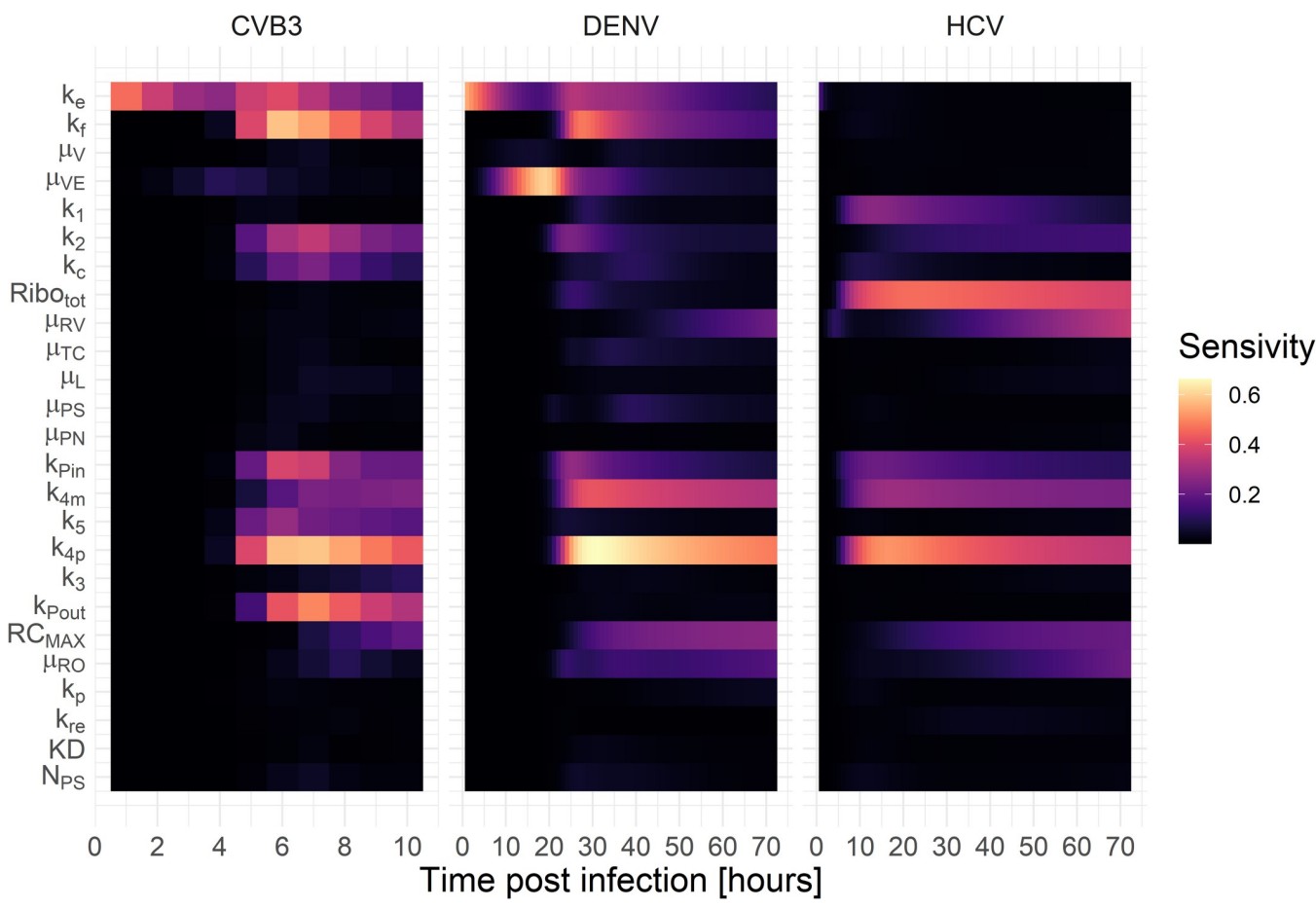

**Fig 6. Global sensitivity profile for the model species plus-strand RNA throughout infection (CVB3 = 10 hours, HCV = DENV = 72 hours).**

For all viruses and drug administration time points, we determined the critical drug efficacy, $\varepsilon$, where the viral life cycle is successfully inhibited and the *in-silico* infection is cleared. Note that we define a virus infection as cleared if the extracellular virus is reduced by more than 99%. By testing both drug administration time points, we found that at the beginning of infection (0 h pi), inhibiting any process led to eradicating the virus (Fig 7). Since the viral replication machinery is not established, viral entry and vRNA release may be possible drug targets. However, an almost 100% inhibition ($\varepsilon$~1) was necessary to block the infection process (S1 Table). Obviously, *in-silico* drugs targeting virus entry and vRNA release at a time point after an established viral infection cannot reduce the viral load. However, for both drug administration time points, targeting vRNA translation and vRNA synthesis showed the most potent effect and, thus, are the most promising drug targets (S1 Table). Interestingly, targeting the formation of the replicase complexes could not clear (or even reduce) CVB3 infection with a drug administration given in steady state (S1 Table). Moreover, in the case of DENV, targeting vRNA export from the RO into the cytoplasm in steady state led to a 6% increase in virus with incomplete inhibition. Only a 100% inhibition and thus a drug efficacy of 1 could clear the virus by 99%.

Since most DAAs are highly efficient in combination, we determined the critical drug efficacy of individual drugs inhibiting either translation complex formation, vRNA translation, or polyprotein cleavage used in combination with drugs that inhibit vRNA synthesis or formation

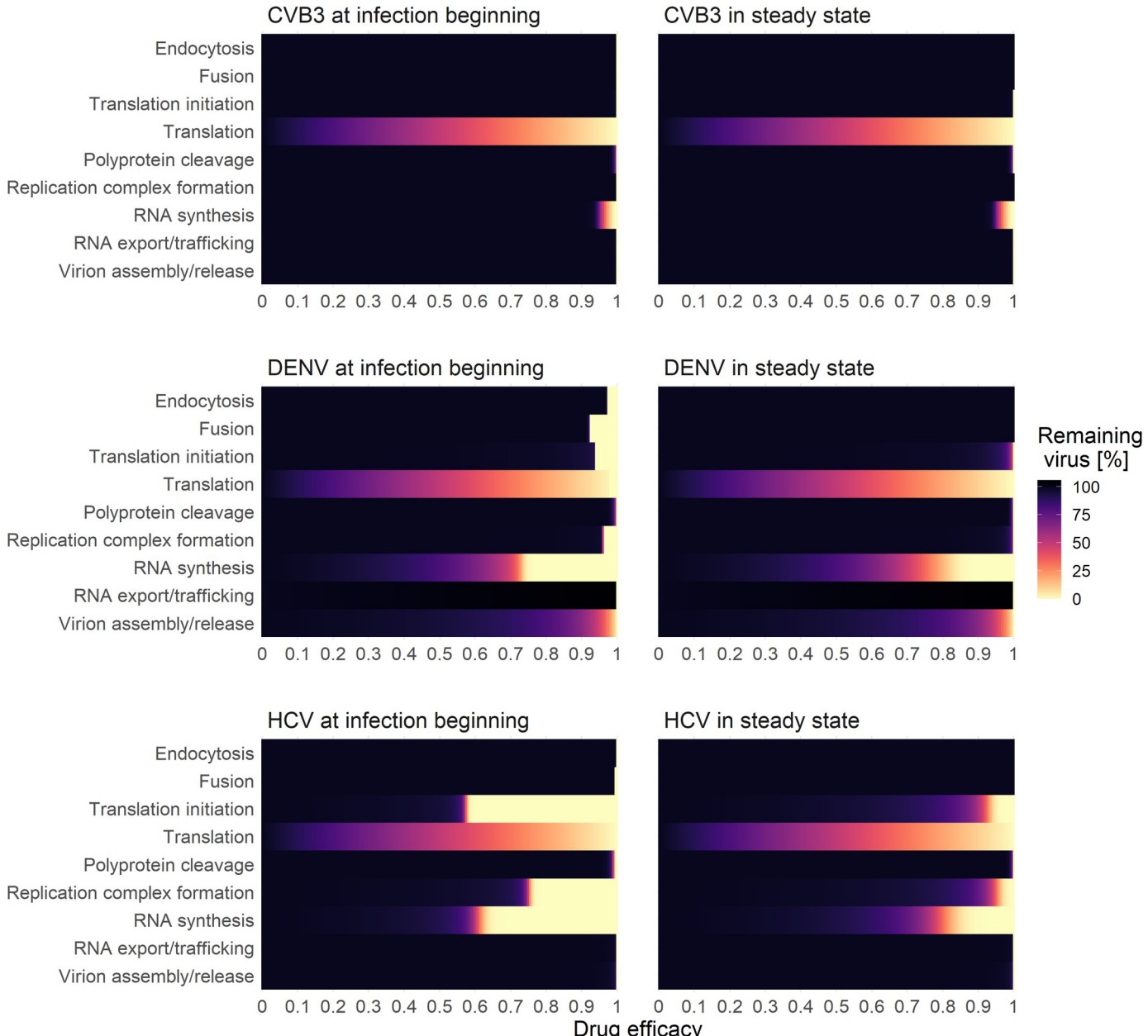

**Fig 7.** Effects of drug interventions applied to two different time points: at infection beginning (left) and in steady state (right). A successful drug treatment leads to more than 99% viral eradication (light yellow), while an ineffective drug treatment leads to 100% remaining virus (black).

of the replicase complex in steady state (Figs 8, 9, S1, and S2 and S1 Table). Here, we identified the "sweet spot" for efficient viral eradication (by more than 99%). Our model predicted that HCV and DENV showed a comparable pattern of viral clearance to a combination of two drugs. In contrast, for the clearance of CVB3, higher drug efficacies were necessary to clear the infection. Inhibiting vRNA synthesis and either vRNA translation or polyprotein cleavage by more than 90% was an efficient combination for HCV and DENV (Figs 8B, 8C, and S2A and S1 Table). However, to clear the infection in all viruses, vRNA synthesis and either translation or polyprotein cleavage must be inhibited by more than 99% or 98%, respectively (Fig 9B and 9C).

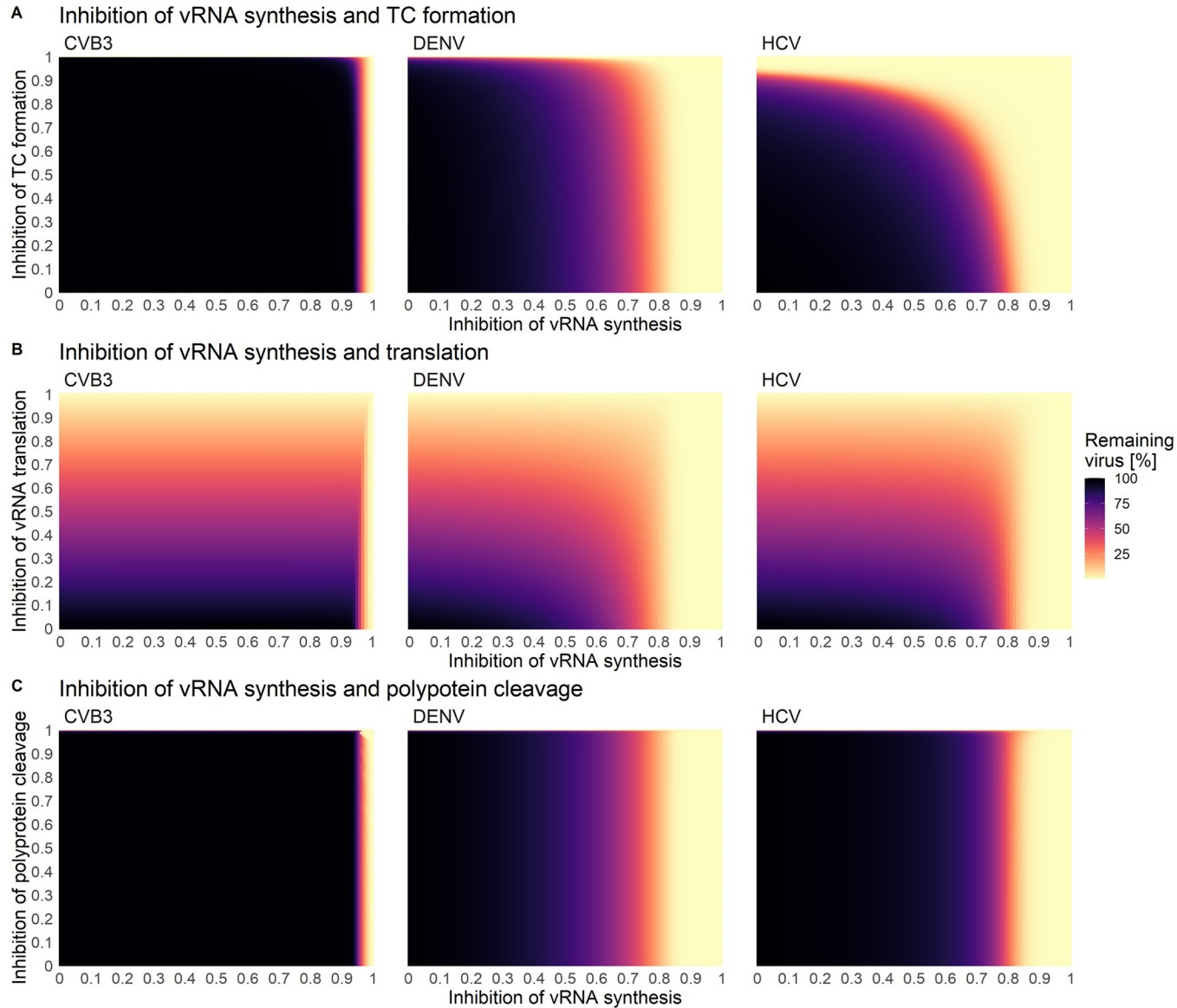

**Fig 8.** Combined drug effect on **A)** vRNA synthesis and formation of translation complex (TC), **B)** vRNA synthesis and translation, and **C)** viral RNA synthesis and polyprotein cleavage. Initiation of treatment was in steady state (100 h pi). A successful drug treatment leads to more than 99% viral eradication (light yellow), while an ineffective drug treatment leads to 100% remaining virus (black).

Interestingly, inhibiting vRNA synthesis and translation complex formation by more than 76% showed the overall lowest critical drug efficacy to clear the infection in HCV. Nevertheless, for CVB3, the vRNA synthesis and translation complex inhibition need to be higher than 99.3% to clear the infection with an almost 10 hours time-delayed viral clearance (Figs 8A and 9A and S1 Table). Overall, we found the lowest pan-viral critical drug efficacy was for the combined inhibition of vRNA synthesis and polyprotein clearance with a required 98% effectiveness for each drug (Figs 8C and 9C and S1 Table). Note that we also tested *in silico* the combination therapy of inhibiting translation complex formation, vRNA translation, and polyprotein cleavage together with replicase complex formation. However, higher critical drug efficacy constants were needed to clear the infection (S1 and S2 Figs and S1 Table).

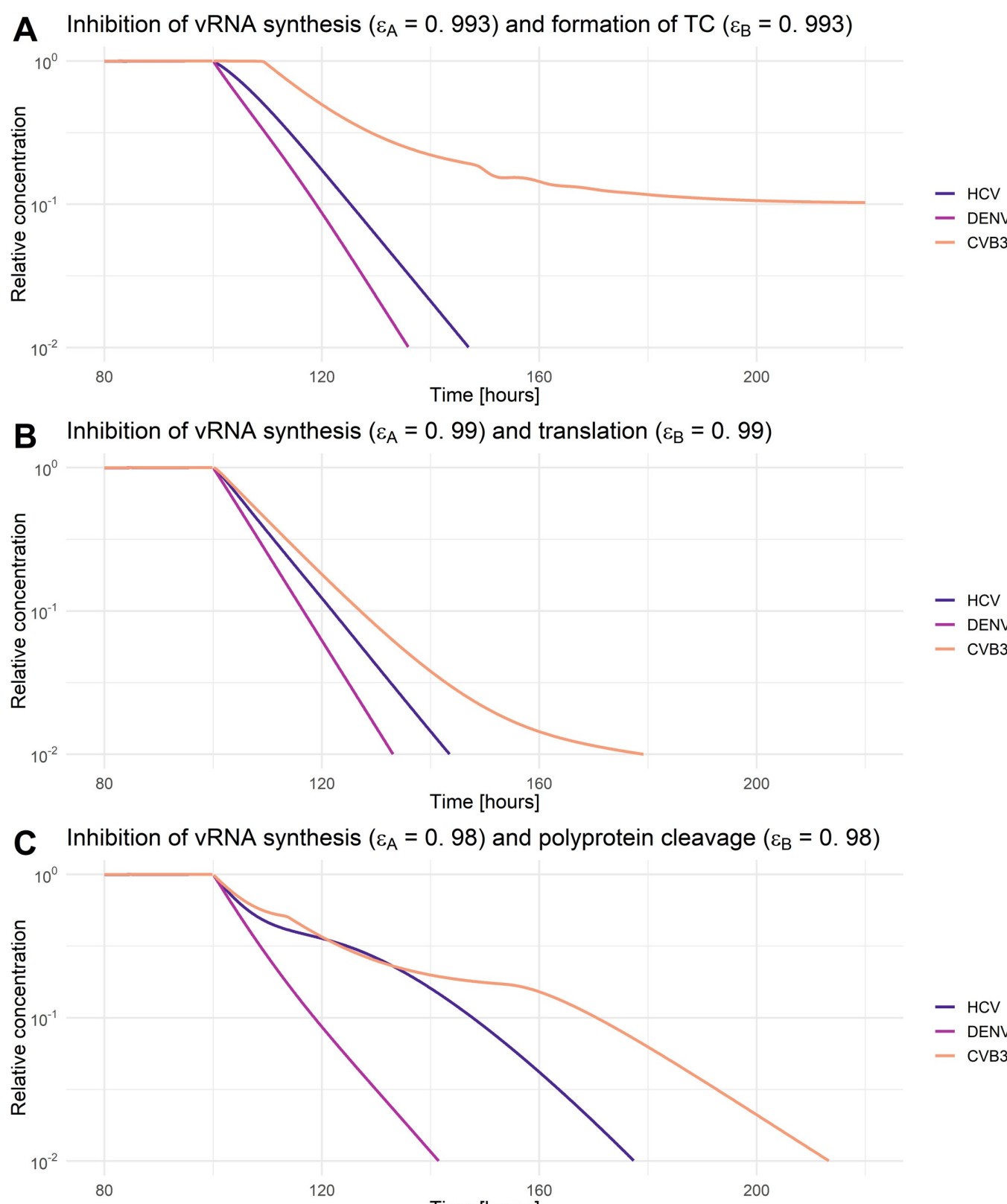

**Fig 9. Relative virus decay under combination therapy that clears HCV, DENV, and CVB3 infections.** A combined drug effect on **A)** vRNA synthesis and formation of translation complex (TC), **B)** vRNA synthesis and translation, and **C)** viral RNA synthesis and polyprotein cleavage. Initiation of treatment was in

steady state (100 h pi). The drug efficacy constant ($\varepsilon_A$ and $\varepsilon_B$) were chosen as minimal efficacies to clear all three viruses. For comparability, virus-specific concentrations in steady state have been normalized to their virus-specific pre-treatment steady-state concentration. A successful drug treatment leads to more than 99% viral eradication (light yellow), while an ineffective drug treatment leads to 100% remaining virus (black).

## Discussion

Mathematical modeling of viral dynamics has a long history and has been applied to various viral infectious diseases [25]. Population-based models considering susceptible and infected cell populations, especially studying virus-host interactions and treatment opportunities for HIV, HCV, and influenza, represent the most prominent mathematical models in the field [25,75–78]. However, mathematical models considering intracellular viral replication mechanisms in detail are still limited and are usually developed for one specific virus such as HCV [19,57,59,79,80], DENV [55], CVB3 [81], HIV [82], or influenza A virus [60,61,83–88]. Furthermore, those virus-specific models are usually developed to study a particular aspect of the viral life cycle, such as cell-line-specific HCV RNA replication efficiency [19] or the life cycles of DENV or CVB3 in the presence of the immune response [55,81]. Recently, Chhajer et al. (2021) studied the viral life cycles of the plus-strand RNA viruses HCV, Japanese encephalitis virus, and poliovirus with a simplified mathematical model. The authors mainly focused on the slow and delayed kinetics of the intracellular formation of replication organelles, which may predict infection outcomes [89].

To our best knowledge, we present here the first mathematical model that simultaneously studies the complexity of intracellular viral replication kinetics for three different representatives of plus-strand RNA viruses, namely HCV, DENV, and CVB3, measured in the same cell line–Huh7. Hepatocyte-derived cells, such as Huh7, support the viral replication of many viruses, such as DENV [90–92], chikungunya [93], Zika [94–96], poliovirus [97], SARS-CoV-2 [98], and other respiratory viruses [99]. The Huh7 cell line can study the viral replication without perturbations of the host cellular immune response due to its defective RIG-I signaling [100]. As we have previously shown that different cell lines lead to different replication kinetics due to a cell-line specific gene expression [19,50,55], our aim was a standardized experimental design, where using the same cell line for all viruses may have the advantage of a mostly shared gene expression and, thus, host factor equality.

The basis for our present study were our previously published intracellular models for HCV [19,57] and DENV [55], which we generalized and adapted to reflect the intracellular replication mechanisms of plus-strand RNA viruses more broadly, as well as the underlying experimental conditions. We compare viral replication mechanisms, pan-viral similarities and virus-specific differences, which may help to understand acute or chronic infection outcomes that may be an initial step toward developing broad-spectrum antiviral treatment strategies.

Our best-fitting model showed high similarity with the virus-specific data and a high degree of parameter identifiability. However, it showed one shortcoming in capturing the dynamics of the experimental measurements of virus in DENV: the viral peak and subsequent drop of the extracellular DENV concentration around 32 h pi. However, our previously published DENV model showed that the dynamics of extracellular infectious virus was dependent on host factors that were packaged into the virions [55]. Since we did not include host factors in the current model, except for ribosomes, we aimed to describe the average extracellular virus dynamics for the first 25 h pi. In the final model, we estimated 31 parameters, of which 27 were identifiable. The 95% confidence intervals of four parameter values hit the upper or lower boundary of estimation, where changing the parameter boundaries by up to 1000-fold did not improve the model fit or improved identifiability.

The non-identifiable rate constant of the naïve cell infection $k_{re}$ may be explained by the fact that reinfection in our culture system may not occur for each virus. However, the process remained in the final model because of different MOI infection experiments, where a lower MOI (MOI of 1, as in the case of CVB3 and HCV) may account for multiple rounds of infection. The formation rate of the translation initiation complex $k_1$ seems to be a non-identifiable process in the model structure, as it was also non-identifiable in our previous DENV model [55]. Further, virus entry and vRNA genome release, $k_e$ and $k_f$, were practically non-identifiable for HCV. An explanation for both processes being non-identifiable may be insufficient experimental measurements for HCV to uniquely estimate both rate constants, e.g., the lack of intracellular protein concentration measurements for HCV. However, since both parameters were identifiable for CVB3 and DENV and both processes were selected as virus-specific, $k_e^{HCV}$ and $k_f^{HCV}$, they remained in the final model as virus-specific. For a detailed comparison of the plus-strand RNA model with our previously published HCV and DENV models, see S3 Text.

## Virus-specific differences and pan-viral similarities

Studying similarities and differences in the viral RNA translation and replication strategies of different viruses is experimentally challenging. Our mathematical model may shed light on this topic by studying 25 processes, from cell infection to releasing newly packaged infectious virions. Five processes within the viral life cycle were determined to be virus-specific: (i) virus entry, (ii) release of vRNA genome, (iii) the number of ribosomes available for vRNA translation, (iv) formation of replicase complexes, and (v) trafficking of newly produced viral genomes from the RO into the cytoplasm.

**Virus internalization and genome release.** The three viruses we studied each have different internalization processes mediated by differences in attachment/entry versus uncoating receptors [101]. HCV replicates *in vivo* in hepatocytes and, consequently, showed the most efficient internalization and genome release processes in our studied hepatocyte-derived Huh7 cells. *In vitro*, HCV replicates most efficiently in Huh7 cells and its closely related sub-clones, while the infection of other cell lines has been challenging [102]. However, both DENV and CVB3 have a broad tropism. DENV infects monocytes, macrophages, and dendritic cells, while CVB3 infects the brain, cardiac tissue, and hepatocytes [15,35,103–105]. Thus, the faster internalization and genome release of CVB3 compared to DENV, and thus its ability to replicate very well in Huh7 cells, is not surprising due to its broader cellular tropism. Nevertheless, DENV RNA has been isolated from various organs and tissues, including the liver (see [106] and references within). However, whether DENV replicates in hepatocytes is under debate [107–109].

**Viral RNA translation.** Among the plus-strand RNA viruses we studied, CVB3 represents the fastest replicating virus with a life cycle of around 8 to 10 hours. Newly synthesized CVB3 RNA is detectable at two h pi in the Golgi apparatus, the site of ROs and thus vRNA synthesis. Levels of viral RNA increase rapidly and peak four h pi [110]. One key feature of successful CVB3 RNA replication is its ability to shut off host mRNA translation, carried out by the virus by degrading eukaryotic initiation factor eIF4G important for the cellular cap-dependent translation complex formation. The result is not only the rapid availability of non-structural proteins required for replicase complex formation [111] but also a lower level of components of the cell's intrinsic immune response. Interestingly, we found the highest total ribosome availability for CVB3, in agreement with its ability to shut off the translation of the host's mRNA while keeping vRNA translation high due to a very efficient IRES. According to our calculated viral RNA translation rate constants, translation is 2 to 3 times faster than HCV and DENV, respectively. It has been shown that the polysome size–the number of ribosomes

bound to a single CVB3 RNA molecule, which translate the viral genome at the same time–is around 30 ribosomes per polysome but changes throughout the CVB3 life cycle; 40 ribosomes per polysome at the beginning of the CVB3 life cycle and 20 ribosomes later in infection [66,112]. Furthermore, Boersma et al. (2020) found that CVB3 translation rates were independent of the host translation shutdown. However, the authors speculated that a host translation shutdown might boost the CVB3 translation at the end of its life cycle, where host cell resources may be limited [113]. Conversely, for DENV, it has been shown that the DENV RNA template is only sparsely loaded with ribosomes and showed a low translation efficiency [114]. Nevertheless, Roth et al. (2017) found that the host's mRNA translation decreases during DENV infection, suggesting that DENV also can repress the host mRNA translation, although not as efficiently as CVB3 [23]. A partial host cell RNA translation shut-off and, consequently, a higher number of ribosomes available for DENV RNA translation is predicted by our model, with DENV having the second-highest predicted ribosome concentration. Interestingly, even though DENV can partially shut down the host's mRNA translation, this suppression seems less efficient compared to the complete CVB3 host shut-off.

**Formation of the replicase complex.** Our model suggests a faster formation of double-membrane vesicles than invaginations, i.e., HCV and CVB3 showed faster replicase complex formation compared to DENV. Compared to DENV and CVB3, HCV showed a 10- and 4-times faster rate of replicase complex formation, respectively. A possible reason may be cell tropism, with hepatocellular-derived Huh7 cells being the cell line of choice for studying HCV. Interestingly, the host mRNA translation shut-off of CVB3 was not associated with a faster supply of non-structural proteins (RdRp) and, thus, faster replicase complex formation. However, host cell translation shut-off may be associated with higher availability and more efficient utilization of viral resources for the formation of replicase complexes, as suggested by our model. CVB3 reached the maximal number of replicase complexes after around 5 h pi, while HCV used 76% less of the possible cell's carrying capacity. However, cell tropism and, thus, a specific set of host factors involved in the process of replication organelle and replicase complex formation may be the crucial factors in this process, as we have shown previously for HCV and DENV [19,55].

**Viral RNA export from the RO into the cytoplasm.** A striking difference between *Flaviviridae* (HCV and DENV) and *Picornaviridae* (CVB3) concerns the parameter values and model sensitivity against changes of the trafficking of newly synthesized vRNA from the RO to the site of translation. For CVB3, our model suggests intra-compartment trafficking is two orders of magnitude slower than HCV and DENV, with a highly significant sensitivity of this parameter against changes. A possible explanation may lie in the involvement of different compartments or cell organelles in vRNA translation and replication. All viruses need proximity to the rough endoplasmic reticulum and its ribosomes for successful vRNA translation; however, they use different cytoplasmic membranes and, thus, different sites for forming their ROs and thus for vRNA synthesis. *Flaviviridae* remodel the rough endoplasmic reticulum, using membrane vesicles or invagination as the site for vRNA translation and synthesis without being exposed to the (possibly damaging) cytoplasmic environment. Melia et al. (2019) found that CVB3 uses the rough endoplasmic reticulum first and the Golgi later in infection, suggesting a high degree of flexibility and adaptation of CVB3 to its environment. To what extent viral replication occurs on either membrane is unknown. However, other studies suggest that Golgi-derived membranes are the primary origin of viral replication [110,115,116]. During CVB3 infection, the Golgi collapsed and was not detectable anymore, suggesting that ROs were Golgi-derived [117]. Regarding efficient viral protein production for virion packaging, CVB3 is not enveloped. It may only need a fraction of the structural proteins that DENV and HCV need for assembly (see S1 Text for details), implying that CVB3 developed strategies

to overcome longer trafficking distances. However, another explanation may be a possible regulation and competition of vRNA translation and virion packaging. Early in infection, vRNA may be used for translation, while later in infection, vRNA may be packaged into virions and thus not available for vRNA translation.

## Hypothetical mechanisms behind acute and chronic infections

The plus-strand RNA viruses studied here share the major steps in their life cycle and replication strategy, but despite these similarities, they show very different clinical manifestations. While HCV has a relatively mild symptomatic phase, it can establish a chronic infection with low-level viral replication over decades that goes mostly undetected by the host's immune response. In contrast, DENV causes a vigorous acute self-limited infection that can become life-threatening. Similarly, CVB3 usually causes an acute infection with flu-like symptoms but can become chronic. The underlying mechanisms for the development of chronic infections are unclear. Our plus-strand RNA virus replication model might help to reveal the differences in the viral dynamics leading to different clinical manifestations.

DENV/Zika virus and CVB3 produce a higher ratio of plus- to minus-strand RNA (20:1) compared to HCV, with a plus- to minus-strand RNA ratio of 3:1 (measured in our data) up to 10:1 (reported in the literature [113,118–124]), which may be HCV-strain or cell line-specific. One may speculate that a higher viral RNA synthesis rate may be responsible for the higher plus- to minus-strand RNA ratio in viruses causing acute infections. However, our calculated vRNA synthesis rates were comparable for HCV and DENV but 50 times lower compared to the CVB3 RNA synthesis rate, possibly due to faster vRNA copying or faster *de novo* initiation of vRNA synthesis. In HCV, studies found an RNA synthesis rate of 150 to 180 nt/min [125,126]. However, the rate of RNA synthesis in DENV is, to our knowledge, unknown. Nevertheless, Tan et al. (1996) found low in vitro polymerase activity for DENV NS5, which is in line with the polymerase activities for West Nile and Kunjin viruses, suggesting that this is a conserved feature of flavivirus polymerases [127] and possibly *Flaviviridae* including HCV.

As for CVB3, it has been shown that the closely related poliovirus synthesizes a single RNA template in 45 to 100 sec [66]. Additionally, it is estimated that between 3 and 10 RdRps are bound to one single PV RNA genome. However, our plus-strand RNA model did not consider the RdRp density bound to one single viral RNA template due to a lack of data for HCV and DENV. According to our model predictions, critical processes for a faster viral life cycle may be a combination of (1) faster viral RNA translation and synthesis rates and/or faster vRNA synthesis initiation, (2) host cell translation shut-off and thus higher ribosome availability for viral RNA translation and at the same time lower ribosome availability for antiviral protein production, (3) and shorter RNA half-lives for intracellular viral RNA (more important in cell lines with intrinsic immune responses or *in vivo*). Interestingly, the potential role of these key processes is in line with the global sensitivity analysis results: All CVB3 replication process rates within the RO show highly significant sensitivities, suggesting that CVB3 strongly depends on an efficient replicative cycle within the RO. Additionally, global sensitivities of vRNA degradation rates in the cytoplasm or within the RO seem rather negligible.

Our model predicted that an optimal usage of viral resources to form replicase complexes within a cell was only realized by DENV and CVB3. Strikingly, HCV only reached 26% of the cell's replicase complex carrying capacity. A possible reason may be a limitation in viral resources to form replicase complexes such as viral RNA or non-structural proteins. Both may be again related to the lower availability of ribosomes for viral protein production in HCV. In contrast, DENV and CVB3 have the advantage of a partial or complete host cell translation shut-off, respectively. However, virus-specific ribosome availability and translation activity

may be related to different translation mechanisms. While HCV and CVB3 have IRESes, i.e., the RNA translation is cap-independent, DENV's translation mechanism is cap-dependent. Furthermore, different IRES types have variations in their structural elements and recruit host factors as regulatory elements, which affects the translation initiation complex and viral RNA translation. Therefore, a higher ribosome availability for vRNA translation may be associated with different translation mechanisms, such as secondary structures and host factors assisting in ribosome binding [128–131]. Furthermore, a higher number of ribosomes available for vRNA translation may be directly associated with a higher production of viral proteins. However, the more ribosomes available for cellular mRNA translation and thus the production of proteins of the immune response, the higher the intracellular degradation of viral components may be, resulting in a limitation in viral resources. Ribosome availability and its control may thus be crucial for viral replication efficiency.

To analyze this aspect further, we asked whether we could make virus production in HCV more efficient or CVB3 less efficient. Increasing the *in-silico* ribosome availability in HCV to that of CVB3 increased the viral load by three orders of magnitude. In contrast, a 50-fold increase in the HCV RNA synthesis rate had no effect on the viral load in steady state due to a limited availability of the viral RNA polymerase in the replication organelle [19]. In contrast, using only 0.07% of ribosomes for CVB3 RNA translation, thus setting the ribosome level to the number of ribosomes used in HCV, decreased the CVB3 viral load by three orders of magnitude. Interestingly, the coronaviruses' non-structural proteins, including those of SARS-CoV-2, target multiple processes in the cellular mRNA translation, causing a host cell translation shut off similar to CVB3 and DENV [132,133]. Therefore, a repression or complete shut-off of the host mRNA translation machinery may be a key feature of acute viral infections.

Comparing *in vivo* viral dynamics with those of *in vitro* experiments is challenging. Nevertheless, we found a comparable pattern of viral dynamics: reported *in vivo* and in our *in vitro* experiments. *In vivo*, HCV showed an exponential growth rate of 2.2 per day [134], while DENV and CVB3 grow twice as fast with a rate of 4.3 and 4.5 per day in human and murine blood, respectively (approximated from [38,44]). However, in murine cardiac tissue, the *in vivo* CVB3 exponential growth rate increases to approximately 14.5 per day [38]. Furthermore, the different exponential growth rates are associated with variations in the peak viral load. At its peak, HCV produces $10^8$ RNA copies per g liver tissue [43], DENV produces 1 to 2 orders of magnitude more virus ($10^9$ to $10^{10}$ RNA copies per mL blood) [44], and CVB3 produces 3 to 4 orders of magnitude more virus ($10^{11}$ to $10^{12}$ RNA copies per g cardiac tissue) compared to HCV [38]. We found a similar pattern in our data, with HCV producing the least amount of virus at its peak (~1 PFU/mL/cell), followed by DENV (~10 PFU/mL/cell) and CVB3 (~200 PFU/mL/cell). Considering the RNA synthesis rates, CVB3 replicates 50- times faster than HCV and DENV.

## Broad-spectrum antivirals?

DAAs are highly specific drugs usually designed to inhibit the function of one specific viral protein. Developing broad-spectrum antiviral drugs is challenging. Nevertheless, we were interested in the possibility of a pan-viral drug treatment option. We, therefore, studied the core processes in the life cycles of our three representatives of plus-strand RNA viruses and administered *in-silico* drugs in mono or combination therapy to identify single drug targets or combinations of drug targets that yield an efficient inhibition of all three viruses.

**Direct-acting antivirals against HCV.** Several DAAs have been developed and approved for HCV and can cure chronic hepatitis C in most patients [135]. DAAs are developed to target one specific protein such as HCV NS3/4A (e.g., first-generation telaprevir or boceprevir and second-/third generation glecaprevir, voxilaprevir and grazoprevir), HCV NS5A (e.g.,

daclatasvir, velpatasvir, ledipasvir), and HCV NS5B (e.g., sofosbuvir and dasabuvir) [136]. Therefore, the DAAs' modes of action and efficacies may be used here to validate the results of our *in-silico* drug intervention study. While DAAs block HCV NS3/4A and intervene with the polyprotein cleavage, HCV NS5A and HCV NS5B inhibitors target the RO formation and vRNA synthesis, respectively [9,59,137]. Our sensitivity and *in-silico* drug analysis suggested high sensitivities for processes associated with HCV RNA replication, which led to an efficient viral reduction by more than 99% with a more than 90% inhibition of the vRNA synthesis rate. Furthermore, our *in-silico* drug analysis predicted that complete HCV NS3/4A inhibition (more than 99.5% polyprotein cleavage inhibition) was necessary to clear the viral load. Combined with inhibiting vRNA synthesis, a combinatory inhibition of more than 90% led to HCV clearance, where viral clearance was mainly driven by inhibiting vRNA synthesis. Our results are in line with current HCV treatment recommendations that focus on a regimen based on a combination of targeting vRNA synthesis alone by inhibiting HCV NS5A and/or NS5B or in combination with HCV NS3/4A with the inhibition of NS5A as the backbone of an efficient HCV treatment regimen, e.g., the combinations of elbasvir (NS5A inhibitor) and grazoprevir (NS3/4A inhibitor), glecaprevir (NS3/4A inhibitor) and pibrentasvir (NS5A inhibitor) or sofosbuvir (NS5B inhibitor) plus velpatasvir (NS5A inhibitor) [138]. Interestingly, the combinatory inhibition of vRNA synthesis and polyprotein cleavage showed pan-viral clearance with the lowest critical efficacies of 0.98, i.e., a 98% inhibition of both processes.

**Broad-spectrum antivirals and host-directed therapy.** The cure of a chronic hepatitis C infection represents a success story for DAAs. However, a subset of HCV patients report treatment failure, severe side effects that impede treatment success, or drug resistance [139]. No successful treatment has been approved for DENV, the most prevalent mosquito-borne viral disease. Furthermore, the vaccine is only recommended for seropositive individuals due to its increased risk of severe disease in seronegative individuals [140]. Moreover, for enteroviruses, such as myocarditis causing CVB3, no antiviral treatment exists to date. Several DAAs targeting CVB3 have been tested in clinical trials but are often associated with the emergence of resistance and, thus, are not recommended [141,142].

Targeting cellular components crucial for successful and efficient viral replication (so-called host dependency factors) may offer a potential treatment option with a high resistance barrier. Additionally, plus-strand RNA viruses still represent a major health concern infecting millions of people worldwide, including the viruses in this current study–HCV, DENV, and CVB3 – and other plus-strand RNA viruses such as chikungunya, Zika, West Nile, Yellow fever, hepatitis A virus as well as the current global pandemic causing SARS-CoV-2. Even though identifying pan-serotype antiviral agents is challenging, a DENV inhibitor has been identified, which has shown high efficacy and pan-serotype activity against all known DENV genotypes and serotypes [143]. Our model may serve as a basis for the development of further virus-specific models as well as pan-viral broad-spectrum antiviral treatment strategies.

Our sensitivity and drug analysis showed that inhibiting translation complex formation, vRNA translation or polyprotein cleavage, and vRNA synthesis represent the most promising pan-viral drug targets. As in the case of HCV, targeting vRNA replication and polyprotein cleavage has been highly successful, however, directly targeting the HCV RNA translation (e.g., the HCV IRES RNA structure) or its complex formation is mainly experimental. Another treatment strategy may be targeting host factors hijacked by the virus and involved in almost every process of the viral life cycle [144]. A limited number of available ribosomes may be a key feature limiting efficient virus production due to suppressed host mRNA translation or complete host cell translation shut-off. However, targeting and thus inhibiting the biological function of ribosomes will be challenging and not beneficial for the host. Nevertheless, two proteins were found to interact with vRNA translation: RACK1 and RPS25. Both proteins may

be hijacked by DENV and promote DENV-mediated cap-independent RNA translation [145]. Additionally, in HCV RACK1 has been shown to inhibit IRES-mediated viral RNA translation and viral replication; in the latter case RACK1 binds to HCV NS5A, which induces the formation of ROs [146,147]. Similar to HCV, CVB3 RNA translation is mediated through an IRES and, thus, RACK1 may be a potential drug target. Furthermore, studying interactions of SARS-CoV-2 proteins with host mRNA identified RACK1 as a binding partner and thus may represent a pan-viral host dependency factor [148].

Interestingly, the very early processes in the viral life cycle, virus entry as well as fusion and release of the vRNA genome, showed significant sensitivities in DENV and CVB3 but were rather negligible in HCV. Further, the release of the viral RNA genome from endosomes showed a higher significant sensitivity compared to viral entry and internalization. Interestingly, cyclophilin A is a host factor involved in the enterovirus A71 (family *Picornaviridae*) fusion/uncoating process and, thus, vRNA release [149,150]. Furthermore, cyclophilin A inhibitors block or successfully decrease viral replication in several plus-strand RNA viruses such as HCV, DENV, West Nile, yellow fever, enteroviral A71, and coronavirus [142,151]. Considering that it is involved in both processes that showed the highest sensitivities, cyclophilin A may represent a promising pan-viral target [142].

The formation of the replicase complexes represented another sensitive pan-viral process. Replicase complexes are associated with membranes of the ROs either within or outside the RO facing the cytosol [152]. Several studies have shown the significance of host factors in RO formation being associated with cell permissiveness and vRNA replication efficiency [17,101,133,144]. For example, Tabata et al. (2021) have shown that the RO biogenesis in HCV and SARS-CoV-2 critically depends on the lipid phosphatidic acid synthesis since inhibiting associated pathways led to an impaired HCV and SARS-CoV-2 RNA replication [153]. However, even though successful in clearing HCV and DENV, in an established infection of a fast-replicating virus such as CVB3, the formation of replicase complexes may not represent an efficient drug target. In steady state, CVB3 replicase complexes are already formed, and the virus cannot be cleared even with a 100% inhibition given for 5 days. Similar results have been found by targeting host factors involved in the formation of replicase complexes of other picornaviruses. Two tested compounds targeting RO formation could not block viral replication, suggesting that viral replication continues if ROs are already formed [154]. Furthermore, targeting host factors involved in RO formation showed lethal cytotoxicity, as in the case of PI4KIIIβ and HCV [155]. Interestingly, inhibiting the host factor PI4KB showed that CVB3 RO formation was delayed and CVB3 RNA replication occurred at the Golgi apparatus [116].

Interestingly, incomplete inhibition of some processes may promote viral growth. Our model predicted that targeting viral export from the RO into the cytoplasm in the DENV life cycle led to a 6% increase in virus. Therefore, low-efficacy drugs may lead to the opposite of the desired outcome. Thus, host-directed therapy may have an enormous potential on the one hand but may result in substantial side effects on the other hand. Identifying host factors with pan-viral activity without lethal toxicity represents a challenge for future research.

### Limitations and outlook

In the current study, we developed the first mathematical model for the intracellular replication of a group of related plus-strand RNA viruses. Even though our model allowed a high degree of parameter identifiability, fit the *in vitro* kinetic data, and is consistent with the current biological knowledge of our studied viruses, there are some weaknesses to consider.

First, our model focuses on a single cell and does not include viral spread. Especially in acute infections with rapidly replicating viruses, viral transmission within organs may be

highly relevant to consider. However, since our model was developed for a single-step growth curve, we neglected viral spread and focused mainly on intracellular replication processes. Virus-specific mechanisms of viral spread from infected to susceptible cells may be interesting to study in the future.

Second, our experiments were performed in the immuno-compromised Huh7 cell line, and we did not consider an intrinsic immune response here. In the future, considering an intrinsic immune response may be an important addition.

Third, even though plus-strand RNA viruses share remarkable similarities in their replication strategy, our model does not consider viruses with more than one open reading frame and ribosomal frameshift. The difference between viruses with one and more open reading frames is the presence of sub-genomic RNA, as in the case of coronaviruses. However, the life cycle of coronaviruses, and in particular SARS-CoV-2, differs from our model by producing non-structural proteins first, followed by viral RNA and sub-genomic RNA synthesis [156]. The sub-genomic RNA is later translated into structural proteins. However, since the core processes of viral non-structural protein production (necessary for vRNA synthesis) and vRNA synthesis are common, we do not think that the presence of sub-genomic RNA would considerably impact our presented results. Adaptation of the model to coronaviruses is an ongoing topic being followed up on in our group.

Fourth, *in vitro* experiments are not a reliable system for an *in vivo* application. Especially our drug treatment study needs experimental validation. However, our model and *in silico* drug analysis showed a high degree of similarity with the knowledge and efficacy of DAAs available for HCV.

Fifth, our model has been developed for a one-step growth experiment and, consequently, a single cycle of virus growth. Thus, our model predictions are short-term and do not study long-term effects.

In summary, in the present study, we measured the *in vitro* kinetics of three representatives of plus-strand RNA viruses: HCV, DENV, and CVB3. We developed a mathematical model of the intracellular plus-strand RNA virus life cycle based on these experimental measurements. In order to study pan-viral similarities and virus-specific differences, the model was fit simultaneously to the *in vitro* measurements, where the best-fit model was selected based on the AIC and model parameter identifiability. According to our model, the viral life cycles of our three plus-strand RNA representatives differ mainly in processes of viral entry and genome release, the availability of ribosomes involved in viral RNA translation, the formation of the replicase complex, and the viral trafficking of newly produced viral RNA. Furthermore, our model predicted that the availability of ribosomes involved in viral RNA translation and, thus, the degree of the host cell translation shut-off may play a key role in acute infection outcome. Interestingly, our modeling predicted that increasing the number of ribosomes available for HCV RNA translation remarkably enhanced the HCV RNA replication efficiency and increased the HCV viral load by three orders of magnitude, a feature we could not achieve by increasing the HCV RNA synthesis rate. Furthermore, our *in-silico* drug analysis found that targeting processes associated with vRNA translation, especially polyprotein cleavage and viral RNA replication, substantially decreased viral load and may represent promising drug targets with broad-spectrum antiviral activity.

## Supporting information

**S1 Text. Pan-viral and virus specific model parameters.**
(DOCX)

**S2 Text. Model selection process.**
(DOCX)

**S3 Text. Comparison of the plus-strand RNA virus replication model with our previous models.**
(DOCX)

**S1 Data. Experimental data and data underlying manuscript figures.**
(XLSX)

**S1 Table. Critical drug efficacy constants in mono and combination therapy and an in-silico drug administration in steady state (100 h pi).** For simplicity, we assume that both drugs have the same efficacy in combination therapy. The lowest critical drug efficacies to clear the virus-specific infection is highlighted in red (TC = translation complex, RC = replicase complex)
(DOCX)

**S1 Fig.** Combined drug effect on **A)** replicase complex (RC) formation and formation of translation complex (TC), **B)** replicase complex (RC) formation and polyprotein cleavage, and **C)** replicase complex (RC) formation and vRNA translation and drug administration in steady state (100 h pi). A successful drug treatment leads to more than 99% viral eradication (light yellow), while an ineffective drug treatment leads to 100% remaining virus (black).
(TIFF)

**S2 Fig. Relative virus decay under combination therapy that clears HCV, DENV, and CVB3 infections.** A combined drug effect on **A)** formation of replicase complex (RC) and formation of translation complex (TC), **B)** formation of replicase complex (RC) and translation, and **C)** formation of replicase complex (RC) and polyprotein cleavage. Initiation of treatment was in steady state (100 h pi). The drug efficacy constant ($\varepsilon_A$ and $\varepsilon_B$) were chosen as minimal efficacies to clear all three viruses. For comparability, virus-specific concentrations in steady state have been normalized to their virus-specific pre-treatment steady-state concentration. A successful drug treatment leads to more than 99% viral eradication (light yellow), while an ineffective drug treatment leads to 100% remaining virus (black) (see S1 Data).
(TIFF)

## Author Contributions

**Conceptualization:** Carolin Zitzmann, Marco Binder, Lars Kaderali.

**Data curation:** Carolin Zitzmann, Christopher Dächert, Bianca Schmid, Hilde van der Schaar, Marco Binder.

**Formal analysis:** Carolin Zitzmann, Lars Kaderali.

**Funding acquisition:** Martijn van Hemert, Frank J. M. van Kuppeveld, Ralf Bartenschlager, Marco Binder, Lars Kaderali.

**Investigation:** Carolin Zitzmann, Christopher Dächert, Bianca Schmid, Hilde van der Schaar, Martijn van Hemert.

**Methodology:** Carolin Zitzmann, Marco Binder, Lars Kaderali.

**Project administration:** Marco Binder, Lars Kaderali.

**Resources:** Martijn van Hemert, Alan S. Perelson, Frank J. M. van Kuppeveld, Ralf Bartenschlager, Marco Binder, Lars Kaderali.

**Software:** Carolin Zitzmann.

**Supervision:** Martijn van Hemert, Alan S. Perelson, Frank J. M. van Kuppeveld, Ralf Bartenschlager, Marco Binder, Lars Kaderali.

**Validation:** Carolin Zitzmann.

**Visualization:** Carolin Zitzmann.

**Writing – original draft:** Carolin Zitzmann, Frank J. M. van Kuppeveld.

**Writing – review & editing:** Christopher Dächert, Martijn van Hemert, Alan S. Perelson, Ralf Bartenschlager, Marco Binder, Lars Kaderali.

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
