## [Decision Letter · Decision Letter 0]

27 Sep 2022

Dear Prof. Dr. Kaderali,

Thank you very much for submitting your manuscript "Mathematical modeling of plus-strand RNA virus replication to identify  broad-spectrum antiviral treatment strategies" for consideration at PLOS Computational Biology.

As with all papers reviewed by the journal, your manuscript was reviewed by members of the editorial board and by several independent reviewers. One reviewer was generally positive about the manuscript, while the other found serious deficiencies (particularly see comments under modeling design/approach).  In light of this, we would like to give you the opportunity to revise the manuscript for PLoS Computational Biology if you so desire.  A revision should include substantial modifications to take into account reviewer #2's concerns.

We cannot make any decision about publication until we have seen the revised manuscript and your response to the reviewers' comments. Your revised manuscript is also likely to be sent to reviewers for further evaluation.

Please prepare and submit your revised manuscript within 60 days. If you should need more time to address the reviewer concerns, please let us know the expected resubmission date by replying to this email. 

Sincerely,

Peter M Kasson

Academic Editor

PLOS Computational Biology

Virginia Pitzer

Section Editor

PLOS Computational Biology

Reviewer's Responses to Questions

**Comments to the Authors:**

Reviewer #1: In this manuscript, the authors present findings from both in vitro experiments and mathematical modeling to provide further understanding on plus-strand RNA viruses. They develop a new model, which describes the infection dynamics of HCV, DENV and CVB3. The authors calibrated this model to experimental data and used it to examine the similarities and differences between these virus species that operate on very different time scales. The computational study of intracellular targets for broad-spectrum antivirals will help guide the development of new antivirals for plus-strand RNA viruses. The work is clear and well written. I have a few comments regarding this manuscript.

Major comments:

1. The experiments for HCV, CVB3 and DENV were performed at MOIs of 1, 1 and 10, respectively. In contrast, the estimated initial virus concentrations show values of 0.2, 1 and 1. Do you think this difference affects the results of your experiments and that your simulations correctly reflect the initially available virus material?

2. In your model, the total ribosome concentration (Ribo_tot) appears to be the most crucial factor influencing the infection dynamics and time scales for the three viruses. Related to this I have four questions:

(a) You note that this total ribosome concentration is only a fraction of the total number of ribosomes per cell specific to each virus. Does this number represent the ability of ribosomes to bind to these viral genomes and the higher and lower values represent some kind of binding affinity or overall availability of these ribosomes?

(b) It seems like this total ribosome concentration induces the large differences in infection time scales in your model (together with the high synthesis rate for CVB3). How reliable would this deduction be and was this already considered previously?

(c) You derive conclusions regarding the suppression of host cell mRNA translation from the model. However, such a mechanism is not implemented in the model itself. Do you base this solely on the values estimated for total ribosome concentration or is there further supporting evidence from your model? Moreover, is this suppression capability the major factor for the ribosome concentration and ultimately induces the different time scales of the viruses?

(d) Why is the ribosome concentration included as the only host factor in the model, was it previously identified as the most crucial factor? Furthermore, why is the actual concentration of ribosomes included and its effect is not represented only in the reaction rates (TC formation and protein translation)?

3. The entry and RNA release rates (k_e and k_f) for the three viruses show large differences. HCV rates are deemed non-identifiable and are estimated to the upper boundary, while the other two have significantly lower rates. These rates lead to the large majority of initially provided virus material of HCV to be available in the cytoplasm after around 0.3 h. However, based on the estimated rates, only a marginal fraction of the infecting viruses for DENV and CVB3 actually reach the cytoplasm (after 10 h, only around 5 and 12% of the total initial virus material, respectively). For these two viruses, this small amount of provided genomic material is amplified via high translation and replication rates. Ultimately, only a fraction of the infecting virus is used for replication and everything released from endosomes after the first minutes/hours is irrelevant. Thus, these low rates, especially for k_f, seem quite unrealistic to me. Can you comment on that?

4. In line 315, it is mentioned that the newly formed double-stranded RNA and the non-structural proteins are released from the RO. However, that is not depicted in the model scheme and the following steps appear to still occur inside of the RO. Do the double-stranded RNA and the non-structural proteins actually leave and re-enter the RO?

Minor comments:

1. Short title: plos-strand  plus-strand

2. Line 77-78: significant burden  significant burdens

3. 137: RNA translation[22]  RNA translation [22]

4. 188: prof. R.  Prof. R.

5. 537: Splitting of two words over the whole line.

6. 552: The abbreviation “IRES“ is introduced in line 690, but already used in lines 552-553.

7. 600: Table,)  Table)

8. 697: shup down  shut down

9. Figure 1: In the model scheme, the replicase complex inside the replication organelle is shown as RC^RO. However, in the model equations it is used as only RC while the other variables have the exact same names as in the model scheme. An adjustment of the denomination either in the scheme or in the equations would increase clarity.

10. Figure 7, 8, S1: On the x-axis, it seems like 0.9 is followed by 10. If possible, separate the figures a bit further to split up the numbers 1 and 0.

11. You have determined which processes are virus-specific by employing the AIC and considering parameter identifiability. Do you think by using a larger set of experimental data to inform the model additional distinctions between pan-viral and virus-specific processes could be made?

12. The experimental data and models were not available at the time of the review and will apparently be made available later. Thus, the results could not be reproduced. Please supply all code, well documented and with instructions, such that a reader could rerun your simulations and reproduce your figures/tables/results.

Reviewer #2: This work aims to develop a generic mathematical model for plus-strand RNA viral replication to identifying antiviral treatment strategies. To do so, the authors modified/simplified their previous published models into the proposed generic model that was calibrated against previous published viral kinetic experimental data of hepatitis C virus (HCV), dengue virus (DENV), or coxsackievirus B3 (CVB3) in cell cultures. The authors suggest, via model selection process against the measured experimental data, that many viral-host parameter were similar among the 3 investigated viruses. Thereafter, to examine antiviral treatment strategies they simulated antiviral perturbations in each virus-specific model at the beginning of infection (time 0) and 100 hr post infection, ie., at steady state. Comments and suggestions follow

Experimental data:

a. It would be important to be clear how the data were obtained (digitized from a figure, performed by the authors, or had access to the raw data)

b. The 3 investigated viruses have different cell tropism (as summarized in Table 1), with DENV not infecting hepatocytes et all so it would be important to explain why Huh7 cells were used. Also, for HCV experiments Lunet-CD81 cells were used so it is not clear whether they are Huh7 cells.

c. DENV experiment was used MOI=10 while in the others it was MOI=1. Is that was accounted in the in silico modeling?

d. How many cells were placed in each well in each experiment? I wonder whether the units FFU/ml and molecules/ml might have differed if the number of cells in each well is different.

Modeling design/approach

While the model selection processes are impressive and provide a decent agreement between the experimental data and the mathematical models, the paper is not an easy to digest where 3 different viruses with different outcomes (acute vs chronic) and cell tropism are compared and discussed. Since models were already developed for HCV by the authors and there are already approved potent drugs for HCV cure, it seems that the generic model could be presented first in the context of HCV. E.g., showing how the generic model agrees with the previously more complex published HCV model, then show HCV-drug perturbation simulations, and if possible, add some experimental data under anti-HCV that should be available in the literature or from the labs of the co-authors of the current paper.

For DNEV, the recent modeling paper that was developed by the authors could be compared with the generic model, at first, bringing confidence. Explain how a virus that does not infect hepatocytes can be investigated in Huh7 cells. Then discuss relevant drug perturbations for DENV and explain why perform these simulations at steady state for a virus that is spontaneously cleared (2-3 weeks infection duration as noted in Table 1).

For CBV3 it would be important for the readers to know which in silico models have been developed (or not) thus far. How the generic model behaves in a very short time (~10 hr of experiment, Fig. 2C) before death of infected cells occurred (which parameters are different compared to HCV and DENV)? How the generic model can be useful if it reaches a steady state contrary to the experimental data (at least beyond 10 hr)? How relevant are the simulations of VBV3 drug perturbations at steady state if the experimental sys does not reach s.t.? These questions are highly important to convince the readers about the overall ambitious approach of the study to develop such a generic model and broadly test antivirals before and after steady state has reached (in the in silico model).

Minor questions/comments

1. Why the in-silico HCV “Virus” curve starts so low (Fig. 2A) compared to DENV and CBV3 shown in figures 2B and 2C.

2. Why to increase number of ribosomes (lines 964-7)?

3. About 30 of model parameter values are not shown (line 351). Would be nice to add a Table for that and the source for the values.

4. Fig. 2: Consider to use different model curves and symbols that could be easily seen using B&W printers.

Suggested study limitations to acknowledge:

1. The model does not account for drug resistance which was a big challenge to overcome in developing antivirals against HCV.

2. Different labs using different viruses may have significant effect on the results.

**Have the authors made all data and (if applicable) computational code underlying the findings in their manuscript fully available?**

Reviewer #1: **No: **The data and code were not provided with the manuscript and their location was left as a blank to fill out later.

Reviewer #2: Yes

PLOS authors have the option to publish the peer review history of their article (what does this mean?). If published, this will include your full peer review and any attached files.

Reviewer #1: No

Reviewer #2: No
---

## [Decision Letter · Decision Letter 1]

26 Jan 2023

Dear Prof. Dr. Kaderali,

Thank you very much for submitting your manuscript "Mathematical modeling of plus-strand RNA virus replication to identify  broad-spectrum antiviral treatment strategies" for consideration at PLOS Computational Biology.

The reviewer with more substantial concerns re-reviewed the manuscript and found it partially responsive to those concerns.  In light of this, we would request that you revise the manuscript to take into account the remaining concerns.  To the editor's reading, these concerns seem primarily to be of clarification and discussion about the context and potential limitations of the study at hand rather than additional modeling.  We look forward to receiving a revised manuscript to address the remaining concerns.

We cannot make any decision about publication until we have seen the revised manuscript and your response to the reviewers' comments. Your revised manuscript may also be sent to reviewers for further evaluation.

Sincerely,

Peter M Kasson

Academic Editor

PLOS Computational Biology

Virginia Pitzer

Section Editor

PLOS Computational Biology

Reviewer's Responses to Questions

**Comments to the Authors:**

Reviewer #2: While some concerns have been addressed, the major ones - pertaining modeling design/approach - were not.

Here are the comments that have not been addressed in the revised manuscript.

While the model selection processes are impressive and provide a decent agreement

between the experimental data and the mathematical models, the paper is not easy

to digest where 3 different viruses with different outcomes (acute vs chronic) and cell

tropism are compared and discussed. Since models were already developed for HCV by

the authors and there are already approved potent drugs for HCV cure, it seems that

the generic model could be presented first in the context of HCV. E.g., showing how the

generic model agrees with the previously more complex published HCV model, then

show HCV-drug perturbation simulations, and if possible, add some experimental data

under anti-HCV that should be available in the literature or from the labs of the coauthors

of the current paper.

For DNEV, the recent modeling paper that was developed by the authors could be

compared with the generic model, at first, bringing confidence. Explain how a virus that

does not infect hepatocytes can be investigated in Huh7 cells. Then discuss relevant

drug perturbations for DENV and explain why performing these simulations at steady

state for a virus that is spontaneously cleared (2-3 weeks infection duration as noted in

Table 1).

For CBV3 it would be important for the readers to know which in silico models have

been developed (or not) thus far. How the generic model behaves in a very short time

(~10 hr of experiment, Fig. 2C) before death of infected cells occurred (which

parameters are different compared to HCV and DENV)? How the generic model can be

useful if it reaches a steady state contrary to the experimental data (at least beyond 10

hr)? How relevant are the simulations of VBV3 drug perturbations at steady state if the experimental sys does not reach s.t.? These questions are highly important to convince

the readers about the overall ambitious approach of the study to develop such a generic

model and broadly test antivirals before and after steady state has reached.

**Have the authors made all data and (if applicable) computational code underlying the findings in their manuscript fully available?**

Reviewer #2: Yes

PLOS authors have the option to publish the peer review history of their article (what does this mean?). If published, this will include your full peer review and any attached files.

Reviewer #2: No
---

## [Editor Report · Decision Letter 2]

9 Mar 2023

Dear Prof. Dr. Kaderali,

Thank you for your revised manuscript and the care with which you answer the reviewer's questions.  I judge that this should be sufficient for publication and thus will editorially accept the manuscript.

We are pleased to inform you that your manuscript 'Mathematical modeling of plus-strand RNA virus replication to identify  broad-spectrum antiviral treatment strategies' has been provisionally accepted for publication in PLOS Computational Biology.

Best regards,

Peter M Kasson

Academic Editor

PLOS Computational Biology

Virginia Pitzer

Section Editor

PLOS Computational Biology

---

## [Editor Report · Acceptance letter]

28 Mar 2023

PCOMPBIOL-D-22-01120R2 

Mathematical modeling of plus-strand RNA virus replication to identify  broad-spectrum antiviral treatment strategies

Dear Dr Kaderali,

I am pleased to inform you that your manuscript has been formally accepted for publication in PLOS Computational Biology. Your manuscript is now with our production department and you will be notified of the publication date in due course.

With kind regards,

Anita Estes
